# Groundwater Quality, Health Risk Assessment, and Source Distribution of Heavy Metals Contamination around Chromite Mines: Application of GIS, Sustainable Groundwater Management, Geostatistics, PCAMLR, and PMF Receptor Model

**DOI:** 10.3390/ijerph20032113

**Published:** 2023-01-24

**Authors:** Abdur Rashid, Muhammad Ayub, Zahid Ullah, Asmat Ali, Tariq Sardar, Javed Iqbal, Xubo Gao, Jochen Bundschuh, Chengcheng Li, Seema Anjum Khattak, Liaqat Ali, Hamed A. El-Serehy, Prashant Kaushik, Sardar Khan

**Affiliations:** 1State Key Laboratory of Biogeology and Environmental Geology, School of Environmental Studies, China University of Geosciences, Wuhan 430074, China; 2National Centre of Excellence in Geology, University of Peshawar, Peshawar 25130, Pakistan; 3Department of Botany, Hazara University, Dhodial P.O. Box 21120, Pakistan; 4Department of Environmental Sciences, Kohat University of Science and Technology, Kohat 26000, Pakistan; 5School of Civil Engineering and Surveying, Faculty of Health, Engineering and Sciences, University of Southern Queensland, West Street, Toowoomba, QLD 4350, Australia; 6Department of Zoology, College of Science, King Saud University, Riyadh l1451, Saudi Arabia; 7Instituto de Conservación y Mejora de la Agrodiversidad Valenciana, Universitat Politècnica de València, 46022 Valencia, Spain; 8Department of Environmental Sciences, University of Peshawar, Peshawar P.O. Box 25120, Pakistan

**Keywords:** heavy metals, geochemical modelling, water quality, sustainable management, health risk, geochemical speciation

## Abstract

Groundwater contamination by heavy metals (HMs) released by weathering and mineral dissolution of granite, gneisses, ultramafic, and basaltic rock composition causes human health concerns worldwide. This paper evaluated the heavy metals (HMs) concentrations and physicochemical variables of groundwater around enriched chromite mines of Malakand, Pakistan, with particular emphasis on water quality, hydro-geochemistry, spatial distribution, geochemical speciation, and human health impacts. To better understand the groundwater hydrogeochemical profile and HMs enrichment, groundwater samples were collected from the mining region (n = 35), non-mining region (n = 20), and chromite mines water (n = 5) and then analyzed using ICPMS (Agilent 7500 ICPMS). The ranges of concentrations in the mining, non-mining, and chromite mines water were 0.02–4.5, 0.02–2.3, and 5.8–6.0 mg/L for CR, 0.4–3.8, 0.05–3.6, and 3.2–5.8 mg/L for Ni, and 0.05–0.8, 0.05–0.8, and 0.6–1.2 mg/L for Mn. Geochemical speciation of groundwater variables such as OH^−^, H^+^, Cr^+2^, Cr^+3^, Cr^+6^, Ni^+2^, Mn^+2^, and Mn^+3^ was assessed by atomic fluorescence spectrometry (AFS). Geochemical speciation determined the mobilization, reactivity, and toxicity of HMs in complex groundwater systems. Groundwater facies showed 45% CaHCO_3_, 30% NaHCO_3_, 23.4% NaCl, and 1.6% Ca-Mg-Cl water types. The noncarcinogenic and carcinogenic risk of HMs outlined via hazard quotient (HQ) and total hazard indices (THI) showed the following order: Ni > Cr > Mn. Thus, the HHRA model suggested that children are more vulnerable to HMs toxicity than adults. Hierarchical agglomerative cluster analysis (HACA) showed three distinct clusters, namely the least, moderately, and severely polluted clusters, which determined the severity of HMs contamination to be 66.67% overall. The PCAMLR and PMF receptor model suggested geogenic (minerals prospects), anthropogenic (industrial waste and chromite mining practices), and mixed (geogenic and anthropogenic) sources for groundwater contamination. The mineral phases of groundwater suggested saturation and undersaturation. Nemerow’s pollution index (NPI) values determined the unsuitability of groundwater for domestic purposes. The EC, turbidity, PO_4_^−3^, Na^+^, Mg^+2^, Ca^+2^, Cr, Ni, and Mn exceeded the guidelines suggested by the World Health Organization (WHO). The HMs contamination and carcinogenic and non-carcinogenic health impacts of HMs showed that the groundwater is extremely unfit for drinking, agriculture, and domestic demands. Therefore, groundwater wells around the mining region need remedial measures. Thus, to overcome the enrichment of HMs in groundwater sources, sustainable management plans are needed to reduce health risks and ensure health safety.

## 1. Introduction

Globally, heavy metals (HMs) such as chromium (Cr), nickel (Ni), and manganese (Mn) contaminating groundwater is a serious problem, due to contamination’s detrimental health impacts [1,2,3]. Mostly, HMs occur in surface water, groundwater, and volcanic dust [3,4,5,6,7] due to the presence of ultramafic and basaltic rocks, which contain enriched Cr up to 200–2400 mg/kg [8,9]. Rock weathering, dissolution of minerals, industrial effluents, and chromite mining release HMs in the water system. Therefore, pollution by HMs such as Cr has become a major environmental problem in Pakistan due to industrialization, urbanization, and chromite mining [10,11]. The quantity and quality of water in Pakistan are declining [12]. Therefore, the annual volume of groundwater assets per person in the last two decades drastically decreased by more than 20% [13]. According to the Food and Agriculture Organization (FAO), every sixth person in the world faces a water shortage [14]. Globally, groundwater contamination by HMs has been considered the utmost serious environmental concern [15,16,17,18,19]. Particularly, groundwater contamination by Cr has enormous health impacts. Cr is considered the 7th most abundant element on earth and 21st in the Earth’s crust [20,21]. Moreover, Cr is one of the core toxic pollutants, 33rd among air toxicants in urban areas, and 7th among the 20th toxic substances registered by the Agency for Toxic Substances and Disease Registry (TS&DR) [22]. In groundwater, Cr is naturally found in two isotopic forms: trivalent Cr (III) and hexavalent Cr (VI) [23,24,25,26,27]. However, Cr (III) serves as an important nutrient [28,29], while Cr (VI) is used for industrial processes [30,31]. The dissolution of Cr is variable in groundwater, showing Cr (III) is insoluble [31], whereas Cr (VI) is soluble [32]. The United States Environmental Protection Agency (USEPA) and World Health Organization (WHO) set guideline values for Cr in groundwater to be 0.1 mg/L and 0.05 mg/L [33,34,35]. 

The geogenic sources of Cr include weathering of rock, erosion, and mineral dissolution [36]. Moreover, anthropogenic factors such as industrial processes, leakage, inadequate industrial waste disposal, poor storage, and chromite mining release Cr in groundwater [3,37]. Cr has been excessively mined by different countries worldwide such as South Africa, Finland, India, Zimbabwe, Philippine, and Kazakhstan [38,39,40,41,42,43]. Cr predominantly occurs as a chromate ore (FeCr_2_O_4_). Moreover, this study is carried out in extensive chromite mines around the ultramafic terrain of the Heroshah complex in Malakand, Northern Pakistan. Groundwater contamination with Cr is extremely dangerous due to its bioavailability, leachability, carcinogenicity, and genotoxicity [44,45]. Mostly, human beings need a specified amount of Cr for normal body growth [46,47]. However, its higher concentration causes toxicity in the form of cancer, kidney, and liver dysfunction [3,48]. Cr is available in different proportions depending upon environmental conditions such as oxidation, reduction processes, environmental setup, groundwater withdrawal, geological settings, and arid and semiarid climatic conditions [49,50,51]. All the above processes play important roles in the release of Cr in groundwater [52,53].

The continuous demand for groundwater supply increases due to industrialization, urbanization, chromite mining, and the growing population. However, migration, transformation, and enrichment of HMs in groundwater systems mostly depend on the structural composition of groundwater aquifers and subsurface soil [54,55]. Moreover, the transformation of HMs in groundwater depends on the composition of aquifer media, types, and concentrations [56]. Microbes such as bacteria and fungi have an effective role in the transformation of HMs from one form into another form [57]. HMs are released into the groundwater system if the HM-retaining capacity of the soil subsurface decreases due to pH fluctuation [58,59]. Many areas of Pakistan are experiencing groundwater deterioration [3,60]. Therefore, effective groundwater management is essential to maintain the long-term survival of groundwater aquifers [61,62]. However, poorly planned land development strategies deteriorate water quality in Pakistan [63]. In Pakistan, the capacity of groundwater resource managers and coordination between concerned stakeholders such as government institutions, communities, and the private sector are inadequate [64,65]. National law allowed local governments to design effective groundwater management plans [66] to address issues including groundwater contamination, wellhead protection, groundwater remediation, over-extraction, overdraft, recharge, improper irrigation, seawater intrusion, water storage, and conservation [67,68,69,70,71]. Groundwater is becoming less available and less safe to use as a result of pollution, over-extraction, and removal techniques [72,73]. Therefore, improved groundwater management is crucial for a healthy and green Pakistan [74].

This study describes the most accurate and practical method for forecasting and visualizing a complicated data matrix to compute the spatial distribution of groundwater pollutants. This technique completely characterized the groundwater data using a combined statistical and geospatial approach [75,76]. It is a valuable tool for fresh groundwater spatial distribution and autocorrelation [77,78,79,80,81]. To interpolate geographical variability, GIS typically uses four techniques: inverse distance weighting (IDW), kriging, spline, and nearest neighborhood [82]. The kriging technique is the most important [83]. Maps of groundwater quality are identified using kriging to categorize the vulnerable areas and determine the missing information representing the vulnerable zones in the groundwater aquifer [48,81,82]. As a result, spatial distribution maps are produced using ordinary kriging and log-normal kriging [84]. To strengthen the scope of this investigation, spatial interpolation and multivariate statistical techniques are used. Positive matrix factorization (PMF), factor analysis (FA), principal component analysis (PCA), multilinear regression (MLR), cluster analysis (CA), and correlation coefficient are all components of multivariate statistics used in this research [85]. In this study, PCA is mixed with MLR, and its findings are compared with the PMF receptor model. These statistical tools aid in locating the origins of pollution in the groundwater system. 

In the above discussion, we described groundwater contamination with HMs in a chromite mining area in Heroshah, Malakand, Pakistan. Particularly, we compared the geochemical profile of chromite mines water with surrounding groundwater in the mining and non-mining areas with the following objectives: (i) to evaluate heavy metals concentrations, geochemical speciation, and hydrogeochemical properties of groundwater; (ii) to compare the geogenic and anthropogenic origins of HMs by using PCAMLR and the PMF receptor model; (iii) to measure human health risk and outline spatial distribution and vulnerability maps using a geostatistical and geographic information system (GIS 10.7). This study is helpful in the evaluation of HMs in groundwater systems around chromite-enriched mines, suggesting possible management tools to reduce groundwater contamination. 

## 2. Study Area

### 2.1. Description of the Study Area

District Malakand is an important area of Khyber Pakhtunkhwa province that lies between 34–35° north and 71–72° east (Figure 1). The present population of the study area is about 720, 295 individuals. About 51% are male and 49% female, and the population lives in two sub-districts: (a) Dargai and (b) Batkhela [86]. Further, the area is divided into rural and urban areas. The population of the rural area is 652,095, and the urban area population is about 68,200. District Malakand is an enriched mineralized area of KPK province and is famous for chromite, mica, and uranium deposits. The study area is bounded by important range Kohi Hindukush on the north side, which includes the mountainous area of district Dir, Swat foothills on the northern side, district Charsadda, and Mohmand melange complex situated on the southern and western side. However, the Mardan region is located on the eastern side. The prime recharge sources of groundwater are the rainfall and snow melting, while the secondary source is the River Swat and its associated canal flowing in the entire plain areas of the region. 

### 2.2. Climate and Hydrology

The hydrological conditions of a region show how local peoples use groundwater resources in various communities. The water sources, such as hand pumps, bore wells, tube wells, dug wells, and springs, are predominantly recharged by precipitation. However, the majority of the water sources are dry during hot summers and severe drought conditions. As a result, the Swat River and its tributaries are crucial in recharging the nearby water sources. The majority of the groundwater in shallow aquifers is used for drinking, household, industrial, and agricultural uses. Moreover, the residents also obtain water from municipal community tube wells throughout the entire region. The climate of the area is typically semiarid; its bimodal rainfall has peaks in the spring and winter seasons, with an average annual precipitation of 950 mm. The research area experiences tremendously hot summers and severe cold in winter. Furthermore, the majority of the water sources are dry during hot summers due to severe arid conditions [87]. The yearly temperature ranges are 18.2 to 36.8 °C during summer and −6 to −16 °C in the winter season [88]. On the other hand, extreme seasonal change in perceived humidity is observed in the study area. Humidity describes the amount of water vapor present in the atmosphere. Humidity increases with rises in temperature, due to which water evaporates into the atmosphere. The weather in the area is hot with more humidity due to higher temperatures and a fast evaporation rate. Moreover, relative humidity calculations indicate that the water molecules in the atmosphere reach about the maximum moisture content in the air. The high and lower levels of humidity are dangerous for young and old people because of their high health risk exposure. 

### 2.3. Local Geological Setting of the Area

The geological formations of the area comprise five different formations (Figure 1). The geological formations are marked as (a) the Devonian and Silurian (DS) formation, (b) the Precambrian metamorphic and sedimentary (PC-MS) formation, (c) the alluvium formation, (d) the intrusive igneous and metamorphic formation (gg), and (e) the mafic intrusive formation (m) (Figure 1).

The first geological formation represents the geological setting of the Paleozoic age over a timespan of 60 million years. The predominating minerals of the Silurian formation include siliceous rock, clay, and calcareous rock, which contain minerals such as limestone, halite, anhydrite, dolomite, and galena. Meanwhile, the Devonian formation comprises sandstone, mudstone, shale, siltstone, and quartzite minerals. Moreover, early Devonian and late Silurian formations are present on the south side of Kashmir and contain abundant quartzite minerals. However, the undivided and inseparable younger rock is found in Malakand and Chitral areas.

The second geological formation is PC rock, which stands for Precambrian rock, in metamorphic and sedimentary forms. The Precambrian period is the first geologic age and occurred 600 million years ago. This geologic age is characterized by different layers of formation of metamorphic and sedimentary rocks. In addition, Precambrian rock contains a fossilized record of plants and animals buried in the underlying soil. The important minerals contained in this formation include silicate and non-silicate minerals. Precambrian rock includes gneiss, granite, and schist and contains deposits of gold, nickel, iron, mica, biotite, fluorite, copper, sillimanite, chromite, quartzite, marble, shale, cordierite, slate, and garnet. 

The third geological formation includes alluvium deposits, comprising loose and unconsolidated substances made of gravel, sand, silt, and clay particles. The most dominant minerals are ilmenite, rutile, monazite, mica, schist, zircon, phlogopite, rutile, quartzite, and dolomite. The soil made of such minerals has an enormous impact on economic development, and mineral resources play a leading role in global attention.

The fourth geological formation is made of igneous intrusive and metamorphic rock composed of gneiss, granite, and schist rock. This geological formation is composed of granite, granodiorite, tuff, pumice, scoria, dacite, obsidian, and diorite. The dominant minerals of the metamorphic rock include muscovite, amphibole, quartz, biotite, feldspar, serpentine, talc, calcite, garnet, epidote, and chlorite. The igneous rock is composed of quartz, feldspars, plagioclase, orthoclase, biotite, and muscovite. However, the fifth geological formation is composed of mafic intrusive rock, signified by (m). The important areas occupied by this formation include Maina, Mousamina, Totai, and Sellay Patty areas. The enriched minerals contained in this area observed during the initial field survey were quartzite and chromite rocks. The minerals that contain mafic intrusive rock include pyroxene, amphibole, olivine, mica, and quartzite. Interestingly, all the geological formations are composed of enriched Ca^+2^, Mg^+2^, Na^+^, K^+^, and Si minerals.

## 3. Methodology

### 3.1. Groundwater Sampling

Fifty-five groundwater and five chromite mines water samples were collected from drinking water sources such as bore wells, dug wells, hand pumps, and tube wells closest to human settlements in the two tehsils (a) Dargai and (b) Batkhela, along with chromite mining regions around Heroshah complex. The chromite mining areas lie within the premises of Tehsil Dargai, while Tehsil Batkhela is the control region for this study. Thus, 35 groundwater samples were collected from the Dargai mining region, 20 from the non-mining region (control area), and 5 from chromite mines water. The groundwater and mine water samples were collected in such a way that the water samples show an equal and uniform representation of the area. Each sample was collected in 100 mL polyethylene bottles at a distance of 600 m by adopting a random sampling strategy. The bottles were rinsed and acid-washed with 10% HNO_3_ for 24 h. The polyethylene bottles were cleaned properly with deionized water. 

Duplicate samples were collected in 100 mL polyethylene bottles. All water sampling points were noted through a handheld Garmin GPS. One set of samples was used for physical parameters analysis, and the second set of water samples was acidified with pure HNO_3_ by using 3–4 drops to bring the pH of samples below 2. The acidification of water for chemical analysis is important to avoid chemical reactions, adsorption, and precipitation of metal ions. However, the acidified water sample was used for major cations such as Na^+^, K^+^, Ca^+2^, and Mg^+2^ and HMs such as Cr, Ni, and Mn. Meanwhile, non-acidified water samples were tested for the analysis of pH, EC, temperature, TDS, turbidity, PO_4_, NO_3_, HCO_3_, Cl, SO_4_, and ORP. A potable water quality checker (WQC) was calibrated before testing pH, TDS, EC, and turbidity in the groundwater. Major anion was tested through a DR-5000 spectrophotometer using a standardized turbidimetric method, in the Geochemistry Lab of the National Excellence Centre in Geology, University of Peshawar. However, HCO_3_ was analyzed by the titrimetric method, and Cl by the Mohar method, whereas major cations and HMs were measured by ICPMS (Agilent 7500 ICPMS). The geochemical speciation of groundwater variables was measured by atomic fluorescence spectrometry (AFS). The detection limits for Cr and Ni were 0.1 and 0.5. The percentage recoveries of PTEs Cr, Ni, and Mn were 95%, 96%, and 97%, respectively. 

### 3.2. Gibbs and Chadha Plotting

The major ions composition in drinking water samples was measured and represented diagrammatically through Gibbs and Chadha’s diagram using Excel 2022. Gibbs plots were calculated by plotting TDS against (Na/Na + Ca) and (Cl/Cl + HCO_3_), whereas the Chadha diagram was statistically plotted from HCO_3_-(Cl-SO_4_) against (Ca + Mg)-(Na + K) using their computation formula through a probabilistic approach. 

### 3.3. Nemarow’s Pollution Index

Nemarow’s pollution indexing (NPI) is an important technique used for estimating the pollution status of groundwater quality. The steps involved in the calculation of NPI values include a concentration of water parameters divided by each parameter’s WHO standard limit. The NPI is used to check the groundwater quality of mining water and non-mining water for 20 parameters. Mathematically, the following formula is used to calculate the NPI.
(1)$$\mathrm{NPI}=\sqrt{\frac{{\left(\left[\left(\frac{1}{n}\right)\sum_{i=I}^{i=n}\;\frac{C_{i}}{S_{i}}\right]\right)}^{2}+{\left(\left[\max\left(\frac{C_{i}}{S_{i}}\right)\right]\right)}^{2}}{2}}$$
where C_n_ is the concentration of the nth parameter of groundwater, and S_n_ is the standard limit of the nth parameter.

### 3.4. Human Health Risk Assessment (HHRA) Model

The groundwater sources were examined for health risk assessment to test the toxicity of groundwater parameters, particularly HMs. Groundwater contaminants can enter the human body through three major routes: dermal, inhalation, and oral intake. Among these routes, the oral intake of groundwater is considered the most promising route of exposure. Therefore, we have calculated HMs exposure through oral intake in the current study based on chronic daily ingestion (CDI _ingestion_), hazard quotient (HQ _ingestion_), and hazard index (HI _ingestion_) equations. However, Cr and Ni are considered carcinogenic, so we calculated the incremental lifetime cancer risk (ILTR) for Cr and Ni. Thus, the toxicity of HMs is dependent on the ingestion rate of groundwater intake. The health risk (HRA) was computed using US-EPA equations [89].
CDI _ingestion_ = C × IR × ED × EF/BW × AT(2)
THQ = CDI/RfD(3)
HI = ∑ HQ _ingestion_(4)

Here, C_HMs_ = HMs concentrations (mg/L), IR = ingestion rate (2.5 L/day for males, 3 L/day for females, and 2 L/day for children), ED = exposure duration (350 days), EF = exposure frequency (365 days/year), BW = body weight (18 kg for child, 65 kg for males, and 62 kg for females), and AT = average time (70 × 365 days). Thus, to evaluate the health risk exposure, all the values of water variables are obtained via the US-EPA database, excluding C_HMs,_ and RfD [90,91]. Consequently, RfD values are taken from [92]. The hazard index (HI) is calculated by summation of HQ values of each HM. Interestingly, HI indices less than one (<1) represent no risk, and indices higher than one (>1) indicate a greater risk for human beings [89,93].
Total hazard indices (THI) = ∑ THQ _ingestion_(5)

Carcinogenic risk in the groundwater consumed by the local people was tested through oral ingestion (see Equation (5)). The carcinogenic risks of Cr and Ni that were measured for the ingestion route had cancer slope factor (CSF) values of 0.5 and 0.91. The USEPA allowable limit of cancer risk, 1.0 × 10^−4^–1.0 × 10^−6^, determines a chance of 1% per 1,000,000 for the people who consumed contaminated water with Cr and Ni for 70 yrs. Thus, the carcinogenic risk higher than 1.0 × 10^−4^ is intolerable [94].
CR = CDI × CSF(6)

### 3.5. Statistical Analysis

The groundwater variables were reported as range, mean, and standard deviation. Statistical comparisons of Cr with groundwater parameters were designed to obtain correlation, and linearity was attained through correlation coefficient (r) and coefficient of determination (R^2^). The groundwater of mining and non-mining regions and chromite mines was statistically analyzed. The principal component analysis, multilinear regression (PCA-MLR), and cluster analysis (CA) were calculated by using IBM SPSS (version 25) and XLSTAT software (2022 version).

### 3.6. Groundwater Mapping, Geochemical Speciation, and Mineral Phases

The groundwater mapping was designed by the geographic information system GIS (version 10.7). The GIS mapping includes interpolation via the Kriging technique, and vulnerability maps represent the pollution hotspot in the region. On the other hand, geochemical speciation was measured by atomic fluorescence spectrometry (AFS), whereas, groundwater mineral phases were calculated by PHREEQC interactive 3.4.0-12927 [95,96]. PHREEQC was used to measure mineral phases for physicochemical variables and HMs such as Cr, Ni, and Mn by following Equation (7).
(7)$$\mathrm{SI}=\log\frac{\mathrm{IAP}}{\mathrm{KT}}$$

### 3.7. Positive Matrix Factorization (PMF) Receptor Model

To comprehend the percentile contribution of contaminated groundwater sources, positive matrix factorization (PMF) employs the factorization algorithm [97]. For source apportionment, the PMF model was recommended by both the United States Environmental Protection Agency (USEPA) and Xiong et al., 2020 [98]. Groundwater concentrations are divided into source profile and factor contribution matrices in the PMF receptor model [99], and the residue matrix is shown with help of the following equations.
(8)$$X_{\mathrm{ij}}=\sum_{k=1}^{p}\left(g_{\mathrm{ik}\;}f_{\mathrm{kj}}+e_{\mathrm{ij}}\right)$$

Here, X_ij_ represents the parameter concentrations for i and j indices of the water sample parameter. g_ik_ shows the concentration of water variable source, f_kj_ is the percentile contribution of water pollution source, representing a model error, and p determines significant factors [100]. The g_ik_ and f_kj_ matrices are optimized by the PMF receptor model to strengthen the x_ij_ matrix. To identify the ideal modification, the PMF model minimizes the value of (Q) as a gauge of modeling quality [101]. The following equation is used to calculate the mathematical value of Q (see Equation (9)).
(9)$$Q=\overset{n}{\underset{i=1}{\sum }}\overset{m}{\underset{j=1}{\sum }}{\left(\frac{e_{\mathrm{ij}}}{u_{\mathrm{ij}}}\right)}^{2}$$

Here, u_ij_ represents uncertainty in the water data; n and m determine the number of water samples and its variable [102]. The mentioned Equations (9)–(11) are useful in calculating the uncertainty for the PMF model.
(10)$$\mathrm{If}\;C>\;\mathrm{MDL},{\;\mathrm{then}\;u}_{\mathrm{ij}}=\frac{5}{6}\times \;\mathrm{MDL}\;$$
(11)$$u_{\mathrm{ij}}=\sqrt{{\mathrm{MDL}}^{2}+{\left(\mathrm{Error}\;\mathrm{fraction}\times C\right)}^{2}}$$

Here, C measures water variable concentration, MDL denotes the maximum detection limit, and the Error fraction determines the error rate, calculated to be 3.5%.

### 3.8. Precision and Accuracy of Groundwater Data

The ions charge balance error of cations and anions (ICBE) was measured and calculated through Equation (8) to determine the accuracy and precision of groundwater data for mining, non-mining, and chromite mines water. All the water samples were found within ±5%.
(12)$$\mathrm{IBCE}=\left(\sum^{\;}\mathrm{Cations}-\sum^{\;}\mathrm{Anions}\right)/\left(\sum^{\;}\mathrm{Cations}\;+\sum^{\;}\mathrm{Anions}\right)\;\times \;100\;$$

## 4. Results and Discussion

### 4.1. Groundwater Composition

Table 1 shows the statistical results of groundwater around the extensive chromite mining region in Malakand, Northern Pakistan. The groundwater had exceeded the WHO recommended limits [103]. The geochemical findings from groundwater of mining and non-mining areas and chromite mines show significant variations. The geochemical profile of groundwater is more enriched than in non-mining regions. 

The range values of HMs Cr, Ni, and Mn’s concentrations in groundwater of the mining region were 0.02–4.5, 0.4–3.8, and 0.05–0.8 mg/L, respectively. Similarly, the aforesaid groundwater variables in the non-mining region were 0.0–2.3, 0.05–3.6, and 0.04–0.6 mg/L, respectively. Likewise, the above-groundwater variables in the chromite mines water were 5.8–6.0, 3.2–5.8, and 0.6–1.2 mg/L, respectively. 

The pH values of groundwater in the mining region and non-mining region were slightly alkaline, while chromite mines water was recorded as slightly acidic-neutral. All the samples were found within the WHO guidelines [103], whereas, the EC values of all groundwater samples exceeded the allowable limit of WHO of 400 µS/cm. The major ions such as Na, Mg, Ca PO_4_^−3^, and HCO_3_^−^ exceeded the WHO guidelines. Moreover, HMs Cr, Ni, and Mn have exceeded the acceptable WHO guidelines values of 0.05, 3.0, and 0.5 mg/L. The percentage contributions of excessive groundwater variables EC, turbidity, TDS, Na^+^, Mg^+2^, Ca^+2^, PO_4_^−3^, HCO_3_^−^, Cr, Ni, and Mn were recorded as 100%, 42%, 20%, 28%, 10%, 13.3%, 100%, 30%, 70%, 30%, and 35%, respectively. The increasing order of cations and anions in groundwater of mining and non-mining areas increases in the following pattern: mining region > non-mining region. The groundwater of the mining region is extremely influenced by granite, gneisses, ultramafic and basaltic rocks, and known active chromite mining around Heroshah, Malakand. All the samples of mine water exceeded the HMs concentration due to water–rock interaction, weathering, and erosion of ultramafic rocks. The results of this study were compared with the findings of Rashid et al., 2021 and 2022, conducted in the neighboring water system of Mardan Basin and the floodplain area of River Swat. The results are consistent and accurate in explaining the environmental condition and parental rock interaction of bedrock materials within the aquifer [3,60]. Overall, the groundwater contamination in the ultramafic terrain of Malakand, Pakistan is further aggravated by chromite mining in the vicinity of groundwater in the mining region.

### 4.2. Gibbs Plotting

A Gibbs scatter diagram compiles the control mechanism of groundwater enrichment in the mining and non-mining regions around the extensive chromite mining region of Malakand, Pakistan. The Gibbs diagram estimates the influences of major geochemical processes such as evaporation dominance, rock-weathering dominance, and precipitation dominance. However, several other factors such as underground aquifer composition, bedrock mineralogy, water transportation, climate conditions, and geochemical composition of groundwater are also important. The findings of the Gibbs plotting determine three major control mechanisms: (1) evaporation, (2) rock weathering, and (3) atmospheric precipitation. Thus, groundwater samples of mining, non-mining, and chromite mines water were plotted simultaneously. The TDS values were plotted versus (Na/Na + Ca) and (Cl/Cl + HCO_3_) (Figure 2). Hence, the variation among the mining water, non-mining water, and chromite mines water was minor regarding TDS and major basic ions except HMs. Thus, all groundwater of mining and non-mining areas and chromite mines water were geochemically influenced by rock-weathering phenomena [104]. Therefore, hydrogeological and regional geological settings play an important role in the enrichment of groundwater systems.

### 4.3. Chadha and Durov Diagrams

Chadha and Durov diagrams are used to compile groundwater type composition that suggests different geochemical processes involved in the formation. However, the Durov diagram showed major composition inferred from geochemical processes (Figure 3a,b). Durov’s diagram compiles anions and cations along with pH and TDS. The findings of Durov’s plot are used to define the processes responsible for groundwater recharge and discharge mechanisms. The trilinear plot on the left side represents the cations such as Na + K, Ca, and Mg, whereas the ternary plot on the top side shows anions such as SO_4_, Cl, and CO_3_ + HCO_3_. However, the projection of ions such as cations and anions developed in the center of the square plot. In the expanded Durov plot, a pH is added in the bottom and TDS to the right side of the ternary plot. Meanwhile the Chadha diagram only required major basic ions to classify groundwater samples into four distinct water types. Thus, 46.6% of water shows CaHCO_3_ water type, 26.6% reveals NaHCO_3_ type, 23.3% shows NaCl type, and 1.6% shows Ca-Mg-Cl type.

Figure 3 describes the processes through which groundwater of the ultramafic terrain around the chromite mining region was recharged by Ca-HCO_3_, Na-HCO_3_,_,_ and NaCl aquifer water types that originate as a result of dissolution and groundwater–rock interaction, weathering of rock, ion exchange, and evaporation processes. Moreover, the water of the recharged areas mixed with existing aquifer water containing albite and calcite minerals leads to the formation of NaHCO_3_, and CaHCO_3_ water types. However, the saline condition and alteration of the groundwater aquifer led to the formation of NaCl water type. Figure 4 displays groundwater samples reported as Ca-HCO_3_, Na-HCO_3_, and NaCl water types, representing a mixed water type that is influenced by rock weathering, ion exchange, and mineral dissolution mechanisms [105,106]. The geochemical results of Figure 4 were compared with the results of Gao et al., 2020 [105], which reflected a similar compositional trend to the results of the current study.

### 4.4. Nemarow’s Pollution Indexing

The NPI values of physicochemical variables in groundwater of mining and non-mining areas and chromite mines water are listed (see Table 2). The NPI values of the groundwater parameters such as EC, turbidity, PO_4_, and Cr in mining, non-mining, and chromite mines water showed potential contribution to groundwater contamination. The average NPI values of the aforesaid parameters were 1.93 ± 0.60, 1.31 ± 0.58, 33.0 ± 19.3, and 20.1 ± 15.0, respectively. The NPI results of the current study for groundwater in the mining and non-mining regions show worse water quality. The NPI results declare that EC, turbidity, PO_4_, and Cr had exceeded the permissible limits of groundwater standards. However, environmental scientists are more interested in finding which individual water parameters are potentially involved in contributing to the deterioration of groundwater quality. Thus, NPI in such cases is crucial in finding answers regarding water quality. The results obtained after the calculation for NPI will be NPI ≤ 1 or NPI > 1. NPI values represent that specific parameters have a significant potential for groundwater pollution. The groundwater results of the present study for NPI were compared with the study conducted by Swati and Umesh. The results are consistent and result-oriented [60,107]. The percentage contributions of NPI for exceeded parameters EC, turbidity, PO_4_, and Cr were recorded up to 100%, 42%, 100%, and 100%, respectively. The increasing order of NPI values shows the following pattern: chromite mines water > groundwater in the mining region > groundwater in the non-mining region.

### 4.5. Health Risk of Exposure to HMs

A health risk assessment model was designed to investigate the people of the mining and non-mining regions of ultramafic terrain in Malakand, Northern Pakistan. The probabilistic approach was applied to examine risk exposure in children aged 1–14 years and adults (15–60 years). Most of the residents claimed that traffic pollution, industrial setup, chromite mining, and commercial actions are responsible for the groundwater contamination and environmental health concerns in the study area. The most prevailing diseases reported by the research team are recorded as follows: gastrointestinal disease, irritability, poor appetite, constipation, fatigue, convulsions, sleep disorder, cramps, learning disabilities, inhibit growth, hearing loss, neurological damage, vomiting, flu, kidney disease, and liver dysfunction.

The range values of ADD and HQ for Cr, Ni, and Mn in the groundwater of mining and non-mining areas and chromite mines water consumed by children, males, and females are compiled in Table 3. The range values of lifetime cancer risk (LCR) for Cr and Ni in groundwater consumed by children were 0.001–0.250 and 0.040–0.222 in the mining region; 0.001–0.125 and 0.002–0.182 in the non-mining region; and 0.319–0.333 and 0.223–0.228 in the chromite mines water, respectively. The range values of the aforesaid variables in males were 0.0001–0.087 and 0.014–0.077 in the mining region; 0.0001–0.043 and 0.001–0.063 in the non-mining region, and 0.111–0.115 and 0.077–0.079 in the chromite mines water, respectively. However, the range values of the aforesaid variables in females were 0.001–0.109 and 0.018–0.097 in the mining region; 0.0001–0.054 and 0.001–0.079 in the non-mining region; and 0.139–0.145 and 0.097–0.099 in the chromite mines water, respectively. The total hazard indexing values assessed for LCR in the mining, non-mining, and chromite mines water consumed by children, males, and females were 0.542–0.561, 0.188–0.194, and 0.436–0.444, respectively. 

The comparative statistics show that groundwater sources of mining regions have more contamination than non-mining region water. However, the residents living on the premises of chromite mines consuming mines water for domestic purposes would face severe health impacts. Our investigation reported that children and females of the chromite endemic region face severe health implications due to HMs, specifically Cr and Ni. Non-carcinogenic risks such as HQ and THI and carcinogenic risks such as LCR and THI of Cr and Ni were evident in the children, males, and females. The increasing trend of non-carcinogenic and carcinogenic risk was recorded as children > females > males, for the entire region. The HMs results of this study were compared with the findings of [108] and [48]. Therefore, these results are found accurate and result-oriented.

### 4.6. Geospatial Distribution and Representation of Pollution Hotspot

Groundwater variables were geospatially analyzed to model the distribution pattern and pollution status (Figure 4). The groundwater variables were plotted in the GIS software 10.7 using the ordinary kriging technique to interpolate variables into a visualized model [109,110,111]. The geochemical maps of physicochemical variables and HMs show a spatial distribution pattern, and the variables and HMS were (a) pH, (b) EC, (c) TDS, (d) temperature, (e) Na, (f) K), (g) Mg, (h) Ca, (i) NO_3_, (j) Cl, (k) HCO_3_, (l) SO_4_, (m) ORP, (n) Cr, (o) Ni, and (p) Mn. The enriched and saturated zones of the geospatial model show mobilization of water contaminants in the groundwater aquifers (Figure 5). The majority of the pollutants determine varying degrees of water contamination such as lowest, low, medium, potentially high, and highest. Thus, groundwater contamination depends on the concentration and enrichment of water variables. The Cr in the groundwater system is geospatially distributed, showing five different classes (Figure 4). Similarly, the Ni is geospatially distributed, and its range values were 0.02–0.45, 0.45–0.88, 0.88–1.32, 1.32–1.75, and 1.75–2.19 mg/L. Mn’s range values were 0.01–0.19, 0.19–0.37, 0.37–0.55, 0.55–0.73, and 0.73–0.91 (Figure 4). Therefore, all the groundwater variables such as physicochemical, major ion, and HMs are presented in five distinct classes. The pH, TDS, K, NO_3_, Cl, HCO_3_, SO_4_, and Ni showed low to moderate contamination. However, EC, Na, Mg, Ca, Cr, and Mn show severe pollution in the area. The Cr and Mn contamination in the groundwater sources, specifically the chromite mining region, pose a severe threat to the local population of the area. 

### 4.7. Groundwater Vulnerability Mapping of HMs 

The groundwater is potentially vulnerable to HMs contamination in the ultramafic terrain of Malakand, Pakistan. The different classes show varying degrees of water contamination. The possible classes obtained after GIS mapping were lowest, low, medium, potentially high, and highest. Therefore, the distribution pattern of HMs such as Cr, Ni, and Mn shows contamination (Figure 5). The contamination is significantly higher in the potentially high and highest classes. Thus, the people in ultramafic terrain around chromite mines water need to take special care regarding their vulnerability to and the toxicity of HMs. The groundwater around the chromite mines should be banned for domestic and agriculture purposes. Thus, residents of the area are advised to take precautionary measures in order to safeguard their lives and groundwater sources for domestic demands. 

### 4.8. Geochemical Speciation of HMs and Groundwater Variables

Table 4 describes the geochemical speciation of groundwater variables such as OH^−^, H^+^, HCO_3_^−^, CO_3_^−2^, Ca^+2^, Cl^−^, K^+^, Mg^+2^, NO_3_^−^, Na^+^, PO_4_^−3^, and SO_4_^−2^ and HMs such as Cr^+2^, Cr^+3^, Cr^+6^, Ni^+2^, Mn^+2^, and Mn^+3^. The concentration ranges of Cr^+2^, Cr^+3^, Cr^+6^, Ni^+2^, Mn^+2^, and Mn^+3^ speciation in groundwater of the mining region were 4.0 × 10^−17^–6.0 × 10^−16^, 5.0 × 10^−5^–3.0 × 10^−3^, 2.0 × 10^−14^–2.0 × 10^−7^, 3.0 × 10^−2^–5.0 × 10^−2^, 7.0 × 10^−3^–6.0 × 10^−1^, and 3.0× 10^−24^–8.0 × 10^−23^, respectively. Similarly, in the non–mining region, the concentration ranges were 1.0 × 10^−17^–1.0 × 10^−16^, 3.0 × 10^−5^–5.0 × 10^−7^, 3.0 × 10^−14^–2.0 × 10^−6^, 1.0 × 10^−3^–4.0 × 10^−2^, 7.0 × 10^−2^–4.0 × 10^−1^, and 3.0 × 10^−23^–5.0 × 10^−23^, respectively. However, the concentration ranges of the aforesaid geochemical species in chromite mines water were 9.0 × 10^−2^–4.0 × 10^−1^, 1.0× 10^−16^–4.0 × 10^−12^, 7.0 × 10^−3^–8.0 × 10^−3^, 2.0 × 10^−1^–8.0 × 10^−1^, and 4.0 × 10^−23^–1.0 × 10^−22^, respectively. In this study, the highest concentrations of soluble ions were recorded for HCO_3_^−^, Cl^−^, Na^+^, SO_4_^−2^, Ca^+2^, Mg^+2^, K^+^, OH^−^, and the HM ions Mn^+2^, Ni^+2^, Cr^+3^, Cr^+6^, Cr^+2^, and Mn^+3^. The increasing order of geochemical species reported in groundwater was as follows: HCO_3_^−^ > Cl^−^ > Na^+^ > SO_4_^−2^ > Ca^+2^ > Mg^+2^ > K^+^ > Mn^+2^ > OH^−^ > H^+^ > Ni^+2^ > Cr^+3^ > Cr^+6^ > Cr^+2^ > Mn^+3^. Moreover, geochemical speciation determines the mobilization, reactivity, and toxicity of HMs in a complex groundwater system, which is governed by alkaline conditions and ionic strength. The geochemical species of the present study were compared with the results compiled by [3] Rashid et al., 2022. The findings of both studies are consistent and result-oriented. In comparing mining, non-mining, and chromite mines water, it appeared that waters are enriched due to water–rock interaction, weathering, and dissolution of minerals occurring in the bedrock material of the parent rock. The geochemical speciation of chromite mine water recorded the highest concentrations in comparison with mining and non-mining water. Thus, groundwater sources of the mining water region are contaminated, as a result of human inputs and chromite mining practices.

### 4.9. Mineral Phases of Groundwater Variables

Table 5 reveals the range values of mineral phases for groundwater variables such as major ions and HMs such as Cr, Ni, and Mn in the groundwater. The groundwater sources of mining and non-mining areas and chromite mines water showed saturation due to mineral dissolution and a higher interaction time with bedrock materials. However, the significant positive mineral phases suggest saturation phase and negated variables determine undersaturation. Thus, SI ≥ 1 indicates that mineral is precipitated, and SI ≤ 1 shows the dissolution of minerals. Meanwhile, HMs speciation establishes a leading role in the bioavailability and toxicity of water contaminants. The toxic effect and hazardous nature of HMs in groundwater ingestion are functions of their speciation rather than the total concentrations of toxic metals. The results of this study were compared with the findings of Rashid et al., 2022, which determined a consistent trend of groundwater saturation from surrounding chromite mines in the ultramafic terrain of the Malakand area [60]. 

### 4.10. Hierarchical Agglomerative Cluster Analysis

Hierarchical agglomerative cluster analysis (HACA) is an important technique to represent a groundwater data set in three clusters (Figure 6). During the formation of the different clusters, the most likely data sets were arranged in the same cluster, indicating that all the samples in the cluster originated from a similar source. Meanwhile, those samples occupying other clusters represent that the data set originated from another source. Therefore, groundwater samples were grouped into three clusters, C1, C2, and C3, representing the least, moderately, and severely polluted clusters. The results of the HACA indicated that cluster C1 contains (n = 12) groundwater samples, C2 contains (n = 8), and C3 contains (n = 40) (Figure 6). Data variance within the cluster and between the clusters were 21.59% and 78.41%, respectively. Mostly, the HACA supported the geochemical findings of PCA analysis.

The class centroid values of cluster C1 for HMs such as Cr, Ni, and Mn were 1.12, 1.0, and 0.17, respectively. In C2, values for groundwater HMs such as Cr, Ni, and Mn were 0.82, 1.03, and 0.14, respectively. Similarly, C3 values for groundwater HMs such as Cr, Ni, and Mn were 3.17, 0.73, and 0.37, respectively. Distance from the class centroid was 0, 280.091 for the least polluted cluster C1; 280.091, 0, and 546.835 for the moderately polluted cluster C2; and 823.920, 546.835, and 0 for the severely polluted cluster C3. Moreover, the groundwater sources of the entire region were significantly influenced by geogenic factors such as granite, gneisses, ultramafic rock, and basaltic rock and anthropogenic inputs such as chromite mining. The groundwater showed Cr contamination with a 70% contribution in the entire region. Overall, the severely polluted cluster C3 represented a 66.67% contribution to groundwater deterioration in the entire region. The groundwater HACA results of the present study were compared to the findings reported by Rashid et al. 2019 in the surrounding water aquifers. The findings of this study are found consistent and result-oriented [48]. 

### 4.11. Pollution Identification

#### 4.11.1. Principal Component Analysis Multilinear Regression Technique (PCAMLR)

Groundwater pollution status was identified by the principal component analysis multi-linear regression (PCAMLR) technique. We used XLSTAT 2022 and IBM SPSS Statistics 25 with normalized groundwater data to identify pollution sources and their percentage abundance. PCA is a very interesting technique to reduce the negated impacts of groundwater contamination, once the contamination source has been identified. In this study, we used PCA and MLR techniques simultaneously to interpret the groundwater pollution results in percentage contribution using the approach of Rashid et al. 2018 [81]. The PCA results compile five significant loading factors showing positive and negative loading according to their correlating R^2^ values, considered effective around R^2^ = 0.5, whereas MLR determined the percentile contribution of each pollution source. The significant loading factors compiled by PCA-MLR were F1, F2, F3, F4 and F5, respectively (Appendix A and Appendix A). The findings of PCA-MLR for fifty-five groundwater samples and five mines water samples were determined for twenty groundwater variables. The factors loading and their correlation coefficient values between the first and second loading factors of groundwater and chromite mines water variables are shown (Figure 7a,b). Overall, the cumulative percentages of groundwater and chromite mines water were 73.31% and 90.74%. The individual factor variances of groundwater based on percentage variance were 26.439, 14.239, 13.563, 10.311, and 8.759, and the individual factor variances of chromite mines water were 42.184, 29.224, and 19.333, respectively. The corresponding eigenvalues of variables for significant factors were 5.288, 2.848, 2.713, 2.062, and 1.752, and the corresponding eigenvalues of chromite mines water variables were 8.437, 5.845, and 3.867, respectively.

The first factor, F1, represented a 26.439, % variability, with eigenvalue of 5.288 (Figure 7a). The significant positive loadings of F1 were pH, EC, TDS, Na, K, HCO_3_, Cr, and Mn, with correlation coefficient (r) values of 0.949, 0.604, 0.559, 0.985, 0.686, 0.900, 0.543, and 0.527. However, the negated groundwater of F1 was Mg, Ca, and Ni, with the corresponding coefficient (r) values of −0.858, −0.909, and −0.534, respectively (Appendix A). Factor F1 displayed the strong geochemical interaction of groundwater with bedrock material, major ions, ionic strength, and water–rock interaction, indicating that the weathering of granite rocks, ion-exchanging capacity, and mineral dissolution play important roles in the abundance of major ions in groundwater systems. The potential groundwater variables of F1 take their origin from geogenic and mineral prospects. Moreover, pH, EC, and TDS take their genesis from the dissolution of minerals, leakage, and the saline condition of water aquifers [60,111,112]. Mostly, major ions such as Na, K, Mg, Ca, HCO_3_^−^, and SO_4_^−2^ also originated from natural sources such as ion exchange, weathering of granite and gneisses, ultramafic rock, groundwater–rock interaction, and dissolution of albite (Na-AlSi_3_O_8_), dolomite (MgCO_3_·CaCO_3_), muscovite (KAl_2_(AlSi_3_O_10_) (F, OH)_2_), carbonate (CO_3_^−2^), calcite (Ca-CO_3_), plagioclase (NaAlSi_3_O_8_-CaAl_2_Si_2_O_8_), and gypsum (CaSO_4_·2H_2_O) minerals [112,113]. On the other hand, HCO_3_^−^ in the groundwater aquifer takes its origin from water movement, water–rock interactions with Ca or Mg-carbonate rocks such as limestone, and dolomite forming bicarbonates [114]. The potential sources of Cr, Ni, and Mn were topsoil, rock weathering, gasoline, silage pile, organic matter, compost, and bedrock composition, which all play leading roles in the enrichment of HMs in groundwater systems [3,48,115,116]. The MLR results of factor F1 showed that geogenic and mineral prospects contributed 50.6% of groundwater contamination (Appendix A).

The loading factor F2 showed 14.239% variability, with the eigenvalue of 2.848. F2 showed significant positive loading for EC, TDS, ORP, and Mn, with correlation coefficient (r) values of 0.849, 0.853, 0.659, and 0.594 (Appendix A and Figure 7a). The positive loading of EC, TDS, ORP, and Mn denotes that these groundwater parameters are interdependent and belong to similar origins. Thus, oxidizing and saline conditions support the enrichment of EC, TDS, and ORP. Moreover, the Mn contamination in the groundwater system takes its origin from the dissolution of Mn_2_(SO_4_)_3_, Mn_3_(PO_4_)_2_, MnCl2:4H_2_O, MnHPO_4_, MnSO_4_, pyrochroite (Mn(OH)_2_), pyrolusite (MnO_2_), and rhodochrosite (MnCO_3_), respectively. Additionally, groundwater contamination was attributed to the dissolution of minerals, leakage, saline condition, industrial effluents, and agrochemical fertilizer. Thus, MLR results for F2 determined mixed sources such as geogenic and anthropogenic inputs, representing a 23.5% contribution (Appendix A). Overall, geogenic factors, oxidizing conditions, and ultramafic rock support the dissolution of minerals in complex groundwater systems [117], as do anthropogenic inputs such as agrochemicals, industrial discharge, electroplating, and coal combustion [60].

Factor F3 revealed 13.563% variability, with eigenvalue of 2.713 (Appendix A and Appendix A). The significant positive loading variables of F3 were elevation, temperature, NO_3_, Cl, and SO_4_, with their coefficient (r) values of −0.498, −0.521, −0.680, 0.749, and 0.762, respectively (Appendix A). Groundwater contamination takes its genesis from agriculture practices, industrial waste, seawater spray, agriculture runoff, and surface water infiltration. The MLR showed that 14.9% of groundwater contamination in the area resulted from agricultural practices (Appendix A). Anthropogenic sources include agriculture runoff and industrial activities’ surface water infiltration [115,116].

Factor F4 showed 10.311% variability, with eigenvalue of 2.062 (Appendix A and Appendix A). The significant loading variables of F4 were depth, temperature, PO_4_, and Cr, with their coefficient (r) values of 0.650, −0.600, −0.620, and 0.514, respectively (Appendix A). Factor F4 showed that Cr causes groundwater contamination in the extensive mining region, taking its origin from existing chromite mines. Additionally, mineral phases of Cr such as Cr(OH)_2_, Cr(OH)_3_, Cr(OH)_3_(am), Cr_2_O_3_, CrCl_2_, CrCl_3_, Cr-metal, and CrO_3_ suggested that the dissolution of Cr-enriched minerals causes groundwater contamination. However, PO_4_ takes its origin from phosphate fertilizers and agriculture practices. The MLR results of F4 reported a 2.9% contribution in the entire region (Appendix A). Overall, chromite mining, minerals dissolution, and ultramafic rock play leading roles in groundwater contamination [118].

Factor F5 showed 8.759% variability, with eigenvalue of 1.752 (Appendix A and Appendix A). The significant positive variables of F5 include turbidity, ORP, and Ni, with corresponding coefficient (r) values of 0.673, 0.622, and 0.536. However, the negated loading variables were EC and TDS, with coefficient (r) values of −0.561, and −0.512 (Table 5). Ni contamination in groundwater takes its origin from weathering of mafic and ultramafic rock, dissolution of minerals, and water–rock interaction. The MLR results of F5 showed mixed anthropogenic and geogenic inputs, accounting for an 8.2% contribution (Appendix A). Overall, mixed sources such as minerals dissolution, mafic rock [119], agrochemicals, industrial waste, electroplating, and coal combustion play important roles in groundwater contamination [120].

The PCA findings for chromite mine water compiled three significant factors, F1, F2, and F3, with a total variability of 90.741 and corresponding eigenvalue 18.149 (Appendix A and Appendix A). The first loading factor F1 revealed a 42.184% variance, with the eigenvalue of 8.437 (Figure 7b). The significant positive loading variables of F1 include pH, elevation, EC, temperature, K, PO_4_, ORP, Cr, and Mn, with their coefficient (r) values of0.531, 0.551, 0.682, 0.874, 0.966, 0.819, 0.939, 0.973, and 0.965. However, negated loading variables were depth, turbidity, NO_3_, HCO_3_, and Ni, with their coefficient (r) values of −0.700, −0.934, −0.503, −0.520, and −0.514, respectively (Appendix A) The chromite mines water contamination originated from extensive chromite mining activities, saline water intrusion, water leakage, weathering of mafic and ultramafic rock, water–rock interaction, and dissolution of minerals [60]. The MLR results of F1 showed geogenic and anthropogenic inputs, accounting for a 75% contribution (Appendix A).

Factor F2 showed 29.224% variance with the eigenvalue of 5.845 (Figure 7b). The significant positive variables of F2 were Na, HCO_3_, Cl, and SO_4_, with their coefficient (r) values of 0.917, 0.639, 0.728, and 0.567. Similarly, the negated loading variables were pH, Mg, Ca, NO_3_, and Ni, with their coefficient (r) values of −0.728, −0.758, −0.653, −0.805, and −0.769, respectively (Appendix A). The mine’s water contamination resulted from ion exchange, weathering of mafic and ultramafic rocks, mineral dissolution, and agriculture runoff [121,122]. However, the negated variables have no significant contribution to the enrichment of water pollution. The MLR results of F2 showed mixed sources accounting for a 17% contribution (Appendix A).

Factor F3 displayed 19.333% variability, with the eigenvalue of 3.867 (Appendix A and Appendix A). The significant loading variables of F3 were depth, EC, TDS, HCO_3_, and Cl, with their coefficient (r) values of 0.673, 0.707, 0.903, −0.633, and 0.553 (Appendix A). The potential mines water variables such as EC and TDS take their genesis from mineral dissolution, water leakage, and saline condition, whereas HCO_3_ originated from the dissolution of calcite and dolomite minerals interacting with mines water. However, Cl has resulted in mine water due to soil weathering, Cl-bearing formations, salt spray deposition, and intrusion of saline water into fresh groundwater. Overall, chromite mines water showed that contamination originated from solid wastes and debris of chromite mines, improper disposal of waste, and residence time [123]. The MLR results of F2 denoted anthropogenic inputs of mineral prospects, accounting for an 8% contribution (Appendix A).

#### 4.11.2. Source Apportionment Using PMF

An EPA-PMF receptor model was used to ascertain the contributions and comprehend the sources of the groundwater characteristics [124]. The signal-to-noise (S/N) ratios ranged from 1.2 to 10.0, demonstrating the outstanding quality of the groundwater data [125]. Three to five factors are tested with 20 base runs and random seed numbers to see which one produced the smallest differences between Q (true) and Q (false). When all of the bootstrap runs have fully converged, the PMF model is deemed true and successful. By increasing the number of factors from 3 to 5, the values of Q (true) and Q (robust) were completely converged (i.e., Q (true)/Q (robust) = 0.92), and R^2^ values ranged from 0.50 to 0.99, arranged in the order of pH (0.50), EC (0.92), TDS (0.90), turbidity (0.99), Na (0.54), K (0.52), Mg (0.75), Ca (0.82), PO_4_ (0.99), NO_3_ (0.50), HCO_3_ (0.58), Cl (0.50), SO_4_ (0.51), Cr (0.99), Ni (0.54), and Mn (0.58), revealing significant factors exactly fit with the PMF model. Figure 8 displays the fingerprinting of all significant factors, illustrating how diverse geochemical processes impact groundwater composition (Figure 8). The relative contributions of the individual components (see Figure 9) and the percentile contribution of the contributing elements are shown (Figure 10). 

Factor F1 compiles the outstanding contribution of pH, EC, TDS, Na, K, NO_3_, HCO_3_, Cl, SO_4_, Ni, and Mn, with percentage contributions of 28.1%, 40.7%, 42.8%, 58.7%, 27.8%, 29.9%, 36.9%, 26.8%, 22.9%, 15.9%, and 22.1%, respectively. Factor F2 includes the potential contribution of pH, EC, TDS, K, Mg, PO_4_, HCO_3_, Cl, SO_4_, Ni, and Mn, with percentage contributions of 45.6%, 27.8%, 27.6%, 42.9%, 67.6%, 78.2%, 46.1%, 30.6%, 45.6%, 46.6%, 72.7%, and 29.3%, respectively. Factor F3 compiles EC, TDS, turbidity, Na, K, HCO_3_, and Cr, with percentage abundances of 11.7%, 13.6%, 17.9%, 20.6%, 10.8%, 13.6%, and 86.7%, respectively. Factor F4 shows pH, EC, TDS, Na, K, PO_4_, NO_3_, Cl, and SO_4_, with percentage contributions of 16.0%, 10.7%, 10.5%, 17.4%, 16.8%, 64.3%, 15.6%, 15.1%, and 18.0%, respectively. Factor F5 represents turbidity, and Mn, with percentage contributions of 73.4%, and 48.5%, respectively. However, F1 is directly linked with geogenic sources, F2 with industrial pollution, F3 with mining practices, F4 with mineral dissolution, and F5 with agriculture pollution. Comparing the PCA-MLR and PMF models clarified the same parameters and indicated the same sources as the PCA’s factors F1, F2, F3, F4, and F5. The PCA-MLR results are thus utilized to validate the PMF receptor model.

### 4.12. Sustainable Groundwater Management

Groundwater includes essential major ions such as Na^+^, K^+^, Mg^++^, and Ca^++^, anions such as PO_4_^−3^, Cl^−^, SO_4_^−2^, and heavy metals such as Cr, Ni, and Mn, which are important for human health and body growth [60,89]. Most of these necessary elements must be consumed in specified amounts as part of our daily nutrition [126,127]. The excessive ingestion of HMs causes health implications due to their persistent, bioavailable, and genotoxic nature. Thus, HMs can readily enter the food chain and, through interactions with terrestrial and aquatic systems, could transport water contaminants to the ecosystem [128]. Groundwater contaminants migrate from one biological system into another, posing severe health concerns to human beings and the ecosystem [129]. However, it is believed that excessive HMs intake has hazardous health implications for the sustainable management of groundwater. The noncarcinogenic and carcinogenic health concerns recorded in the present study mean that children are more vulnerable to HMs than adults. Water quality measures such as NPI values compiled the contamination level of HMs, and associated water variables recorded moderate to severe pollution in the entire region. The groundwater contamination problem is extremely worse in those countries where alternate sources of water are insufficient, which could increase the reputation of groundwater for drinking and domestic needs. Hence, it is very important to monitor and protect the current groundwater resources by preventing additional water resources degradation.

Our findings strongly recommend groundwater management and governmental entities take steps to set up a groundwater system monitoring network. This can help with the ongoing monitoring of groundwater availability and quality as well as designing remedial measures to stop groundwater degradation. Local governments ought to advocate for stricter regulations to create safe-drinking-water wells. Additionally, suitable campaigns should be carried out to increase public awareness of the significance of sustainable and safe groundwater resource use.

## 5. Conclusions

The groundwater sources of Malakand, Northern Pakistan, around active chromite mines, exceeded the WHO permissible limits for EC, turbidity, PO_4_^−3^, Na^+^, Mg^+2^, Ca^+2^, Cr, Ni, and Mn by up to 100%, 42%, 20%, 28%, 10%, 13.3%, 100%, 30%, 70%, 30%, and 35%, respectively. The NPI values revealed that EC, turbidity, PO_4_, and Cr represented worse water quality, and their percentile abundances were 100%, 42%, 100%, and 100%. Groundwater HMs contamination was more noticeable in the mining region as compared to the non-mining region due to weathering of granite, gneissic, mafic, ultramafic, and basaltic rock and known active chromite mines. The HHRA model of HMs showed the following increasing order: Ni > Cr > Mn. The lifetime cancer risk (LCR) for Cr and Ni revealed the following pattern: children > females > males. The HACA model showed that groundwater data variance within and between the clusters were 21.59% and 78.41%. The HACA supported the geochemical findings regarding groundwater resulting from mineral prospects and weathering of granite, mafic, and ultramafic rocks. The PCAMLR and PMF receptor model suggested common sources of contamination such as geogenic factors, industrial pollution, mining practices, mineral dissolution, and agriculture pollution. The PCA-MLR results are thus utilized to validate the PMF receptor model. The mineral phases reflected supersaturation and undersaturation due to mineral dissolution and precipitation. Thus, SI ≥ 1 recorded the mineral precipitation, and SI ≤ 1 showed the mineral dissolution. The vulnerability maps showed the lowest, low, medium, potentially high, and highest classes for groundwater pollutants. The people of the endemic chromite area are recommended to take special care regarding their vulnerability to and the toxicity of HM to safeguard their lives. The groundwater of the mining region around the chromite mines water should be properly treated before consumption for domestic and agricultural practices.

## Figures and Tables

**Figure 1 ijerph-20-02113-f001:**
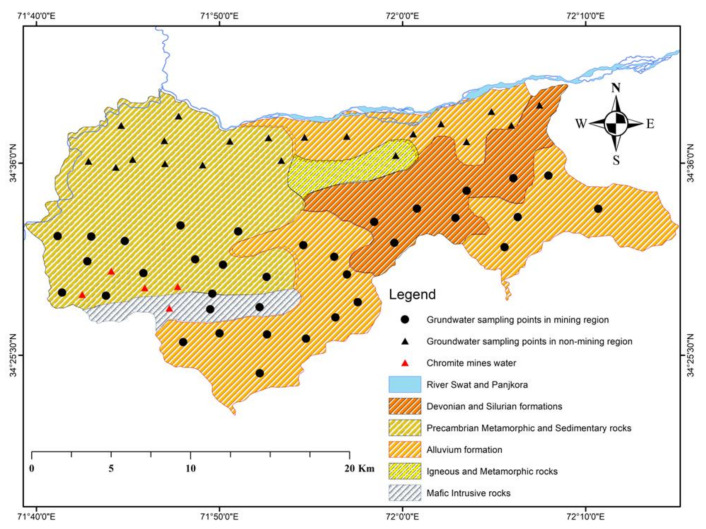
Map showing groundwater samples and geological formations of the area.

**Figure 2 ijerph-20-02113-f002:**
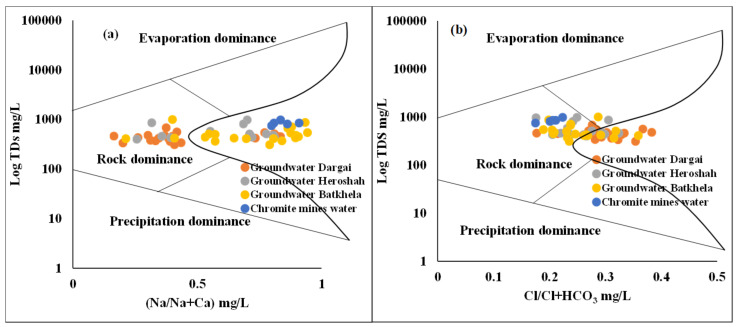
Gibbs plot representing major ion chemistry controlling the groundwater sources of the ultramafic terrain of Malakand, Northern Pakistan: (**a**) Na/Na + Ca and (**b**) Cl/Cl + HCO_3_ mg/L versus Log TDS.

**Figure 3 ijerph-20-02113-f003:**
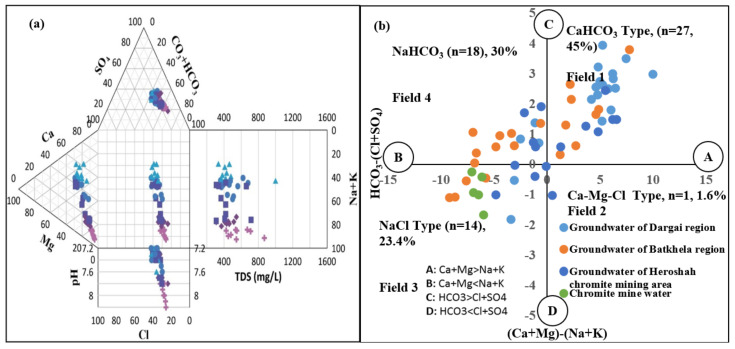
(**a**) represents the Durov plot compiled groundwater of mining with green circle, and non-mining water with blue circle, pH with square, and TDS with plus sign; and (**b**) reveals the Chadha diagram compiling mechanism responsible for groundwater type formations in the study area.

**Figure 4 ijerph-20-02113-f004:**
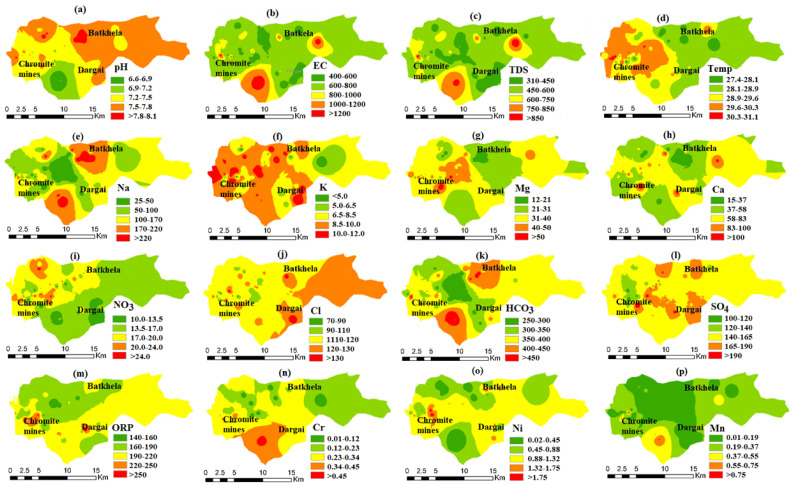
Spatial distribution pattern of physicochemical variables and HMs such as (**a**) pH, (**b**) EC, (**c**) TDS, (**d**) temperature, (**e**) Na, (**f**) K, (**g**) Mg, (**h**) Ca, (**i**) NO_3_), (**j**) Cl, (**k**) HCO_3_, (**l**) SO_4_, (**m**) ORP, (**n**) Cr, (**o**) Ni, and (**p**) Mn, in groundwater system of the study area.

**Figure 5 ijerph-20-02113-f005:**
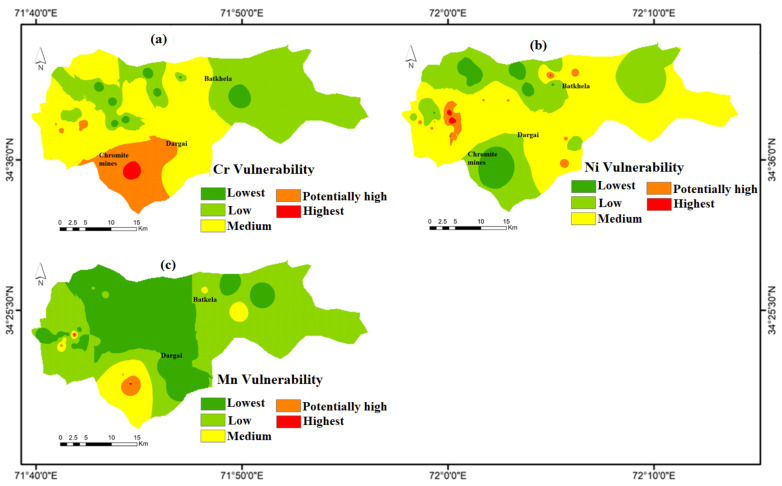
Represents vulnerability maps of HMs such as (**a**) Cr, (**b**) Ni, and (**c**) Mn, in groundwater sources around chromite mines in the ultramafic terrain of Malakand, Northern Pakistan.

**Figure 6 ijerph-20-02113-f006:**
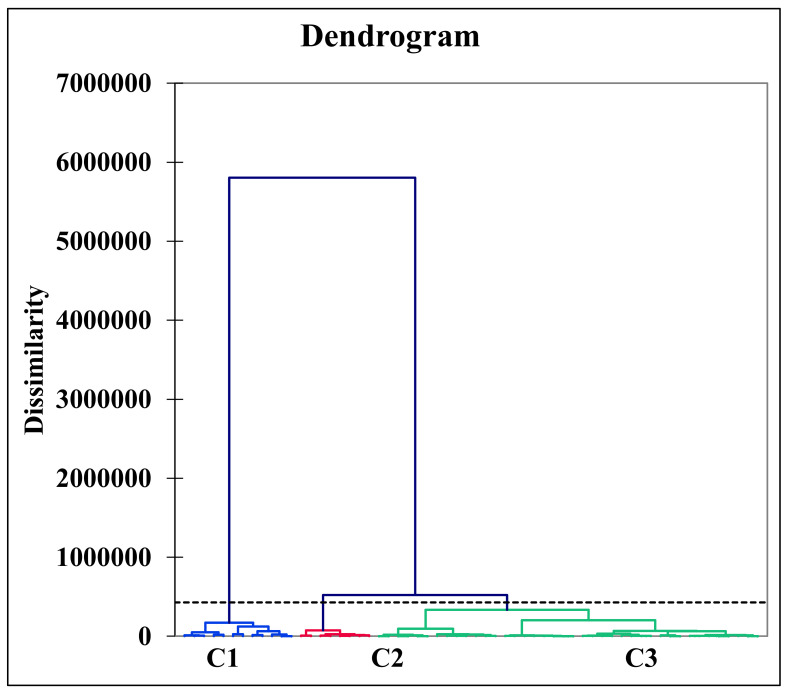
Clustering analysis showing the distribution of groundwater samples after Ward’s classification into three distinct clusters C1, C2, and C3, in the ultramafic terrain of Malakand, Northern Pakistan.

**Figure 7 ijerph-20-02113-f007:**
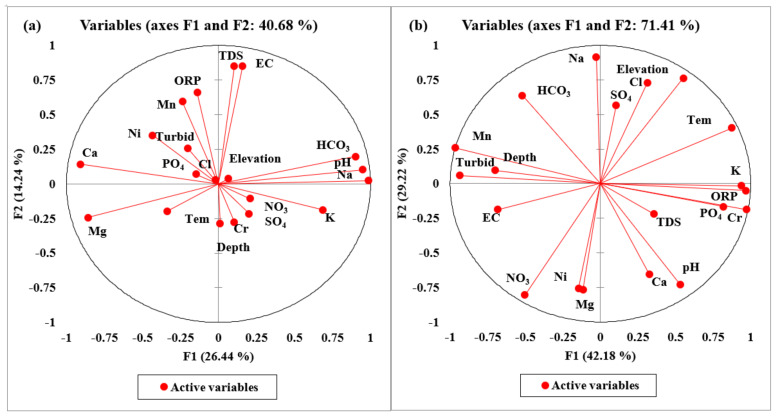
(**a**) Overall loading factors of PCA, and (**b**) relationship of F1 and F2 after varimax rotation in groundwater variables of ultramafic terrain of Malakand, Northern Pakistan.

**Figure 8 ijerph-20-02113-f008:**
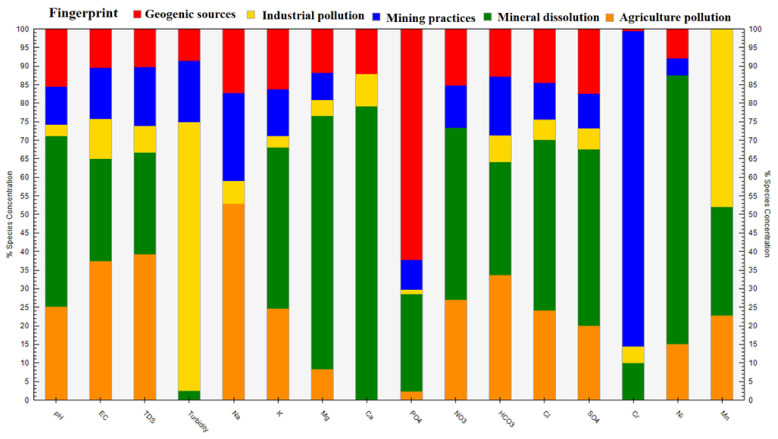
Fingerprints show the distribution of pollution sources in groundwater of the ultramafic terrain of Malakand, Northern Pakistan.

**Figure 9 ijerph-20-02113-f009:**
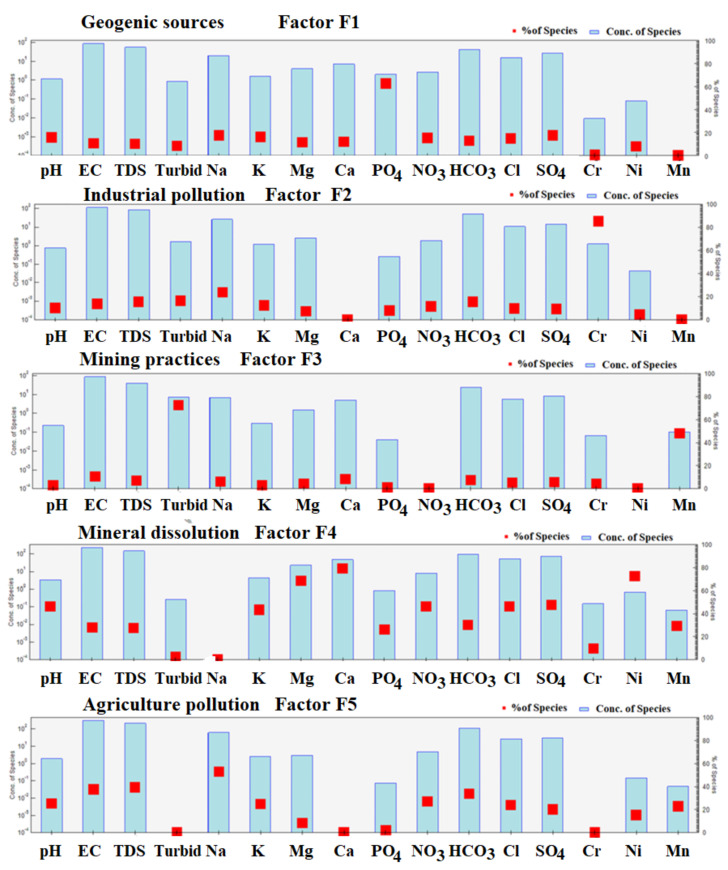
Factors determine the contribution of different processes controlling the groundwater chemistry using the PMF model.

**Figure 10 ijerph-20-02113-f010:**
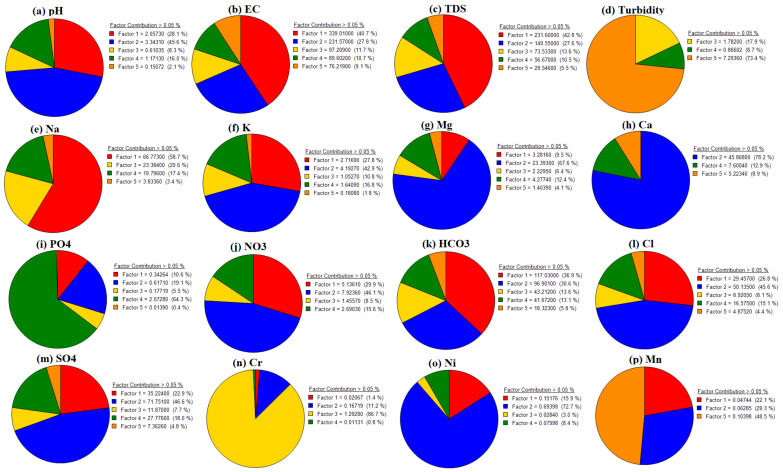
Shows percentage contributions and cumulative sum of individual parameters compiling factors by PMF receptor model.

**Table 1 ijerph-20-02113-t001:** Statistical summary of groundwater around extensive active chromite mines in the ultramafic terrain in Northern Pakistan.

Water Sources	Water in Mine Region (n = 35)		Water Control Area (n = 20)		Chromite Mines Water (n = 5)		
Statistics	Range	Mean ± SD	Range	Mean ± SD	Range	Mean ± SD	WHO Limit
pH	7.2–8.0	7.4 ± 0.2	7.3–8.2	7.71 ± 0.26	6.5–7.2	6.8 ± 0.25	6.5–9.2
Depth m	30–135	74.0 ± 25.5	27.0–95.0	64.6 ± 18.84	15.0–25.0	20.0 ± 3.81	-
Elevation m	290–495	376 ± 57.8	310–500	347 ± 23.92	430–470	452.0 ± 16.43	-
EC µS/cm	480–1500	782 ± 249.1	480–1400	760 ± 233.31	1200–1500	1370 ± 139.6	400
Temperature °C	27.5–31.2	30 ± 1.0	27.4–30.2	28.6 ± 0.76	27.5–31.2	29.2 ± 1.50	-
Turbidity NTU	2.4–14	6.0 ± 2.7	3.2–12.0	6.7 ± 2.83	33.0–70.0	47.6 ± 16.47	5.0
ORP mV	140–280	196 ± 36.5	160–225	191 ± 19	170–260	212.0 ± 38.34	-
TDS mg/L	310–980	503 ± 164.3	310.0–1000	512.5 ± 174.8	750.0–980.0	848.0 ± 84.38	1000
Na mg/L	25–210	86 ± 53.1	30.0–245	152.3 ± 71.05	180.0–255.0	213.0 ± 27.75	200
K mg/L	6.0–12.0	10 ± 1.8	6.0–12.0	9.6 ± 2.01	8.0–12.0	10.6 ± 1.95	12
Mg mg/L	25–60	40 ± 8.5	12.0–45.0	27.7 ± 9.90	21.0–30.0	25.2 ± 3.19	50
Ca mg/L	25–125	66 ± 25.5	15.0–110	49.8 ± 27.65	25.0–50.0	39.0 ± 9.62	100
PO_4_ mg/L	1.33–8.5	3.0 ± 1.9	1.5–10.0	3.5 ± 1.93	1.5–6.5	3.7 ± 1.92	0.1
NO_3_ mg/L	10–28	18 ± 4.7	10–27	17 ± 4	10.0–15.0	12.8 ± 2.17	50
HCO_3_ mg/L	200–650	293 ± 70.1	215–580	350 ± 74	400–630	451.0 ± 30.90	500
Cl mg/L	75–145	110 ± 14.1	95–145	115 ± 13	100–125	115.0 ± 9.35	250
SO_4_ mg/L	100–210	153 ± 27.2	140–190	161 ± 15	140–180	159.0 ± 14.75	500
Cr mg/L	0.02–4.5	1.3 ± 1.2	0.02–2.3	0.7 ± 0.81	5.8–6.0	5.9 ± 0.11	0.05
Ni mg/L	0.4–3.8	1.4 ± 0.28	0.05–3.6	1.2 ± 0.23	3.2–5.8	4.2 ± 0.52	3.0
Mn mg/L	0.05–0.8	0.4 ± 0.12	0.04–0.6	0.3 ± 0.10	0.6–1.2	0.8 ± 0.21	0.5

**Table 2 ijerph-20-02113-t002:** Nemarow’s pollution index (NPI) shows the pollution status of groundwater in mining and non-mining regions within the premises of active chromite mines in the ultramafic terrain of Malakand, Northern Pakistan.

Groundwater Variables	Groundwater in the Mining Region (n = 35)		Groundwater in the Non-Mining Region (n = 20)		Chromite Mines Water (n = 5)	
Statistic	Range	Mean ± SD	Range	Mean ± SD	Range	Mean ± SD
pH	0.92–1.02	0.95 ± 0.03	0.93–1.04	0.98 ± 0.03	0.83–0.92	0.87 ± 0.03
EC µS/cm	1.20–3.75	1.96 ± 0.62	1.20–3.50	1.90 ± 0.58	3.00–3.75	3.43 ± 0.35
Turbidity NTU	0.48–2.80	1.29 ± 0.55	0.64–2.40	1.33 ± 0.57	6.60–14.00	9.52 ± 3.29
TDS mg/L	0.31–0.98	0.50 ± 0.16	0.31–1.00	0.51 ± 0.17	0.75–0.98	0.85 ± 0.08
Na mg/L	0.13–1.05	0.43 ± 0.27	0.15–1.23	0.76 ± 0.36	0.90–1.28	1.07 ± 0.14
K mg/L	0.50–1.00	0.83 ± 0.15	0.50–1.00	0.80 ± 0.17	0.67–1.00	0.88 ± 0.16
Mg mg/L	0.50–1.20	0.80 ± 0.17	0.24–0.90	0.55 ± 0.20	0.42–0.60	0.50 ± 0.06
Ca mg/L	0.25–1.25	0.66 ± 0.25	0.15–1.10	0.50 ± 0.28	0.25–0.50	0.39 ± 0.10
PO_4_ mg/L	13.3–85.0	30.0 ± 19.3	15.0–100.0	35.0 ± 19.3	15.0–65.0	37.0 ± 19.2
NO_3_ mg/L	0.20–0.56	0.37 ± 0.09	0.20–0.55	0.35 ± 0.08	0.20–0.30	0.26 ± 0.04
HCO_3_ mg/L	0.40–1.30	0.62 ± 0.20	0.43–1.10	0.71 ± 0.17	1.02–1.26	1.11 ± 0.09
Cl mg/L	0.30–0.58	0.44 ± 0.06	0.38–0.58	0.46 ± 0.05	0.40–0.50	0.46 ± 0.04
SO_4_ mg/L	0.20–0.42	0.31 ± 0.05	0.28–0.38	0.32 ± 0.03	0.28–0.36	0.32 ± 0.03
Cr mg/L	0.4–90.0	26.9 ± 20.8	0.2–45.0	13.3 ± 10.3	115.0–120.0	116.0 ± 2.2
Ni mg/L	0.13–0.73	0.39 ± 0.13	0.01–0.60	0.25 ± 0.18	0.73–0.75	0.74 ± 0.11
Mn mg/L	0.02–1.60	0.36 ± 0.31	0.22–1.00	0.36 ± 0.20	0.56–2.06	1.14 ± 0.74

**Table 3 ijerph-20-02113-t003:** Compilation of the non-carcinogenic and carcinogenic risks of consuming HMs such as Cr, Ni, and Mn via oral ingestion of groundwater from mining, non-mining, and chromite mines water in the ultramafic terrain of Malakand, Northern Pakistan.

	CDI Children		CDI Males		CDI Females	
	Groundwater in the mining region					
	Range	Mean ± SD	Range	Mean ± SD	Range	Mean ± SD
Cr	0.002–0.500	0.150 ± 0.130	0.001–0.173	0.052 ± 0.045	0.001–0.218	0.065 ± 0.057
Ni	0.044–0.244	0.130 ± 0.043	0.015–0.085	0.045 ± 0.015	0.019–0.106	0.057 ± 0.019
Mn	0.001–0.089	0.020 ± 0.017	0.0004–0.031	0.007 ± 0.006	0.0005–0.039	0.009 ± 0.007
	Groundwater in non-mining region					
Cr	0.001–0.250	0.074 ± 0.088	0.001–0.087	0.026 ± 0.031	0.001–0.218	0.065 ± 0.057
Ni	0.002–0.200	0.084 ± 0.057	0.001–0.069	0.029 ± 0.020	0.015–0.101	0.057 ± 0.019
Mn	0.012–0.056	0.020 ± 0.011	0.004–0.019	0.007 ± 0.004	0.0001–0.039	0.009 ± 0.007
	Chromite mines water					
Cr	0.639–0.667	0.644 ± 0.011	0.221–0.231	0.223 ± 0.004	0.278–0.290	0.281 ± 0.005
Ni	0.246–0.250	0.248 ± 0.002	0.085–0.087	0.086 ± 0.001	0.107–0.109	0.108 ± 0.001
Mn	0.031–0.114	0.063 ± 0.037	0.011–0.040	0.022 ± 0.013	0.014–0.050	0.027 ± 0.016
	HQ Children		HQ Males		HQ Females	
	Groundwater in the mining region					
Cr	0.001–0.333	0.100 ± 0.087	0.001–0.115	0.035 ± 0.030	0.001–0.145	0.043 ± 0.038
Ni	2.222–12.222	6.502 ± 2.151	0.769–4.231	2.251 ± 0.744	0.968–5.323	2.832 ± 0.937
Mn	0.008–0.635	0.142 ± 0.123	0.003–0.220	0.049 ± 0.042	0.003–0.276	0.062 ± 0.053
	Groundwater in non-mining region					
Cr	0.001–0.167	0.049 ± 0.059	0.0001–0.058	0.017 ± 0.020	0.0001–0.073	0.022 ± 0.026
Ni	0.111–10.000	4.181 ± 2.852	0.038–3.462	1.447 ± 0.987	0.048–4.355	1.821 ± 1.242
Mn	0.087–0.397	0.143 ± 0.077	0.030–0.137	0.050 ± 0.027	0.038–0.173	0.062 ± 0.033
	Chromite mines water					
Cr	0.426–0.444	0.430 ± 0.007	0.147–0.154	0.149 ± 0.003	0.185–0.194	0.187 ± 0.003
Ni	12.278–12.500	12.422 ± 0.097	4.250–4.327	4.300 ± 0.034	5.347–5.444	5.410 ± 0.042
Mn	0.222–0.817	0.451 ± 0.262	0.077–0.283	0.156 ± 0.091	0.097–0.356	0.196 ± 0.114
THI	12.926–13.761	13.303 ± 0.366	4.474–4.764	4.605 ± 0.128	13.629–15.994	14.793 ± 0.159
	Carcinogenic risk					
	Groundwater in the mining region					
Cr	0.001–0.250	0.075 ± 0.065	0.0001–0.087	0.026 ± 0.023	0.000–0.109	0.033 ± 0.028
Ni	0.040–0.222	0.118 ± 0.039	0.014–0.077	0.041 ± 0.014	0.018–0.097	0.052 ± 0.017
	Groundwater in non-mining region					
Cr	0.001–0.125	0.037 ± 0.044	0.0001–0.043	0.013 ± 0.015	0.0001–0.054	0.016 ± 0.019
Ni	0.002–0.182	0.076 ± 0.052	0.001–0.063	0.026 ± 0.018	0.001–0.079	0.033 ± 0.023
	Chromite mines water					
Cr	0.319–0.333	0.322 ± 0.006	0.111–0.115	0.112 ± 0.002	0.139–0.145	0.140 ± 0.002
Ni	0.223–0.228	0.226 ± 0.002	0.077–0.079	0.078 ± 0.001	0.097–0.099	0.098 ± 0.001
THI	0.542–0.561	0.548 ± 0.008	0.188–0.194	0.190 ± 0.003	0.436–0.444	0.432 ± 0.003

**Table 4 ijerph-20-02113-t004:** Chemical speciation in groundwater sources of mining and non-mining region and their comparison with chromite mine water in the ultramafic terrain of Malakand, Northern Pakistan.

	Groundwater of Mining Region (n = 35)		Groundwater of Non-Mining Region (n = 20)		Chromite Mines Water (n = 5)	
Statistic	Range	Mean ± SD	Range	Mean ± SD	Range	Mean ± SD
OH^−^	2.0 × 10^−4^–2.0 × 10^−3^	9.0 × 10^−4^ ± 1.0 × 10^−3^	3.0 × 10^−4^–3.0 × 10^−3^	1.0 × 10^−3^ ± 2.0 × 10^−3^	5.0 × 10^−5^–3.0 × 10^−4^	2.0 × 10^−4^ ± 1.0 × 10^−4^
H^+^	1.0 × 10^−5^–9.0 × 10^−5^	5.0 × 10^−5^ ± 6.0 × 10^−5^	8.0 × 10^−6^–7.0 × 10^−5^	4.0 × 10^−5^ ± 5.0 × 10^−5^	8.0 × 10^−5^–4.0 × 10^−4^	3.0 × 10^−4^ ± 2.0 × 10^−4^
HCO_3_^−^	0.184–0.354	0.269 ± 0.1207	0.205–0.315	0.26 ± 0.078	0.372–0.419	0.396 ± 0.034
CO_3_^−2^	3.0 × 10^−1^–0.004	0.002 ± 0.0023	5.0 × 10^−1^–0.005	0.003 ± 0.001	1.0 × 10^−1^–7.0 × 10^−1^	4.0 × 10^−1^ ± 4.0 × 10^−1^
Ca^+2^	0.007–0.021	0.014 ± 0.0094	0.003–0.019	0.011 ± 0.008	0.005–0.008	0.007 ± 0.002
Cl^−^	0.075–0.145	0.11 ± 0.0495	0.095–0.145	0.12 ± 0.035	0.1–0.125	0.113 ± 0.018
K^+^	0.005–0.01	0.007 ± 0.0031	0.005–0.01	0.007 ± 0.003	0.007–0.009	0.008 ± 0.002
Mg^+2^	0.009–0.013	0.011 ± 0.003	0.003–0.01	0.007 ± 0.005	0.006–0.007	0.006 ± 8.0 × 10^−1^
NO_3_^−^	0.01–0.027	0.018 ± 0.012	0.01–0.026	0.018 ± 0.012	0.01–0.015	0.012 ± 0.003
Na^+^	0.021–0.159	0.09 ± 0.0971	0.024–0.187	0.106 ± 0.116	0.142–0.192	0.167 ± 0.036
PO_4_^−3^	2.0 × 10^−5^–3.0 × 10^−4^	3.0 × 10^−4^ ± 4.0 × 10^−4^	3.0 × 10^−5^–7.0 × 10^−4^	3.0 × 10^−4^ ± 4.0 × 10^−4^	2.0 × 10^−6^–5.0 × 10^−5^	3.0 × 10^−5^ ± 4.0 × 10^−5^
SO_4_^−2^	0.076–0.104	0.09 ± 0.0196	0.097–0.12	0.109 ± 0.016	0.096–0.107	0.101 ± 0.007
Cr^+2^	4.0× 10^−17^–6.0 × 10^−16^	3.0 × 10^−16^ ± 4.0 × 10^−16^	1.0× 10^−17^–1.0 × 10^−16^	6.0 × 10^−17^ ± 7.0 × 10^−17^	2.0× 10^−14^–1.0 × 10^−13^	7.0 × 10^−14^ ± 7.0 × 10^−14^
Cr^+3^	5.0 × 10^−5^–3.0 × 10^−3^	2.0 × 10^−3^ ± 3.0 × 10^−3^	3.0 × 10^−5^–5.0 × 10^−7^	3.0 × 10^−4^ ± 4.0 × 10^−4^	9.0 × 10^−2^–4.0 × 10^−1^	2.0 × 10^−1^ ± 2.0 × 10^−1^
Cr^+6^	2.0 × 10^−14^–2.0 × 10^−7^	1.0 × 10^−7^ ± 2.0 × 10^−7^	3.0 × 10^−14^–2.0 × 10^−6^	1.0 × 10^−6^ ± 2.0 × 10^−6^	1.0× 10^−16^–4.0 × 10^−12^	2.0 × 10^−12^ ± 2.0 × 10^−12^
Ni^+2^	3.0 × 10^−2^–5.0 × 10^−2^	4.0 × 10^−2^ ± 5.0 × 10^−2^	1.0 × 10^−3^–4.0 × 10^−2^	2.0 × 10^−2^ ± 3.0 × 10^−2^	7.0 × 10^−3^–8.0 × 10^−3^	8.0 × 10^−3^ ± 7.0 × 10^−4^
Mn^+2^	7.0 × 10^−3^–6.0 × 10^−1^	3.0 × 10^−1^ ± 0.0004	7.0 × 10^−2^–4.0 × 10^−1^	2.0 × 10^−1^ ± 3.0 × 10^−1^	2.0 × 10^−1^–8.0 × 10^−1^	5.0 × 10^−1^ ± 4.0 × 10^−1^
Mn^+3^	3.0× 10^−24^–8.0 × 10^−23^	4.0 × 10^−23^ ± 5.0 × 10^−23^	3.0×10^−23^–5.0×10^−23^	4.0 × 10^−23^ ± 2.0 × 10^−23^	4.0× 10^−23^–1.0 × 10^−22^	7.0 × 10^−23^ ± 4.0 × 10^−23^

**Table 5 ijerph-20-02113-t005:** Description of mineral phases of groundwater in the mining and non-mining regions and their comparison with chromite mines water in the ultramafic terrain of Malakand, Northern Pakistan.

Locations	Groundwater of Mining Region (n = 35)			Groundwater of Non-Mining Region (n = 20)		Chromite Mines Water (n = 5)		
Statistic	Range		Mean ± SD	Range	Mean ± SD	Rang	Mean ± SD	Formula
Anhydrite	0.03–0.79		0.41 ± 0.21	0.27–0.74	0.25 ± 0.28	0.05–0.32	0.17 ± 0.12	CaSO_4_
Aragonite	1.78–2.80		2.23 ± 0.25	2.11–2.58	2.38 ± 0.13	1.07–1.94	1.53 ± 0.31	CaCO_3_
Ca_3_(PO_4_)_2_(beta)	2.63–6.20		4.55 ± 0.84	3.76–5.80	4.78 ± 0.61	0.40–3.94	2.48 ± 1.32	Ca_3_(PO_4_)_2_
Ca_4_H(PO_4_)_3_:3H_2_O	1.74–6.45		4.29 ± 1.15	2.91–5.90	4.44 ± 0.94	−0.78–3.70	1.93 ± 1.70	Ca_4_H(PO_4_)_3_:3H_2_O
CaCrO_4_	−16.86–−10.92		−14.68 ± 1.54	−16.52–−10.28	−13.62 ± 1.61	−20.55–−15.46	−18.26 ± 1.83	CaCrO_4_
CaHPO_4_	0.24–1.49		0.86 ± 0.35	0.29–1.24	0.79 ± 0.34	−0.04–0.89	0.59 ± 0.39	CaHPO_4_
CaHPO_4_:2H_2_O	−0.05–1.21		0.57 ± 0.35	0.02–0.95	0.50 ± 0.34	−0.34–0.60	0.30 ± 0.39	CaHPO_4_:2H_2_O
Calcite	1.96–2.98		2.41 ± 0.25	2.29–2.76	2.56 ± 0.13	1.25–2.12	1.71 ± 0.31	CaCO_3_
Gypsum	0.27–1.03		0.64 ± 0.21	−0.03–0.98	0.48 ± 0.28	0.23–0.56	0.40 ± 0.12	CaSO_4_:2H_2_O
Huntite	4.17–7.78		5.43 ± 0.91	4.46–6.82	6.00 ± 0.67	1.19–4.28	2.73 ± 1.10	CaMg_3_(CO_3_)_4_
Hydroxylapatite	10.77–16.80		14.01 ± 1.38	12.98–16.12	14.52 ± 0.92	6.61–12.76	10.13 ± 2.26	Ca_5_(PO_4_)_3_OH
Lime	−20.93–−19.31		−20.25 ± 0.41	−20.42–−19.46	−19.89 ± 0.26	−22.59–−20.92	−21.79 ± 0.60	CaO
Portlandite	−11.04–−9.43		−10.36 ± 0.41	−10.53–−9.57	−10.00 ± 0.26	−12.70–−11.03	−11.90 ± 0.60	Ca(OH)_2_
Dolomite	4.35–5.93		4.86 ± 0.46	4.46–5.52	5.16 ± 0.30	2.67–4.29	3.50 ± 0.57	CaMg(CO_3_)_2_
Artinite	−4.30–−1.69		−3.41 ± 0.61	−4.04–−2.17	−2.92 ± 0.53	−6.6–−4.37	−5.60 ± 0.81	MgCO_3_:Mg(OH)_2_:3H_2_O
Brucite	−4.98–−3.41		−4.48 ± 0.39	−4.82–−3.57	−4.13 ± 0.35	−6.69–−5.16	−6.00 ± 0.55	Mg(OH)_2_
Epsomite	−2.26–−1.71		−1.95 ± 0.13	−2.50–−1.81	−2.13 ± 0.18	−2.25–−2.11	−2.18 ± 0.05	MgSO_4_:7H_2_O
Hydromagnesite	−5.62–0.15		−3.51 ± 1.31	−5.00–−1.29	−2.61 ± 1.10	−9.85–−5.34	−7.71 ± 1.60	Mg_5_(CO_3_)_4_(OH)_2_:4H_2_O
Magnesite	0.90–1.96		1.30 ± 0.23	1.02–1.67	1.44 ± 0.19	0.28–1.02	0.64 ± 0.26	MgCO_3_
Mg_3_(PO_4_)_2_	−2.32–−0.05		−1.34 ± 0.64	−2.03–0.13	−1.15 ± 0.52	−5.09–−1.95	−3.33 ± 1.16	Mg_3_(PO_4_)_2_
MgCr_2_O_4_	0.06–6.70		3.60 ± 1.69	0.33–7.08	3.62 ± 1.87	−0.12–4.05	1.76 ± 1.50	MgCr_2_O_4_
MgCrO_4_	−24.66–−18.60		−22.41 ± 1.55	−24.41–−17.88	−21.36 ± 1.71	−28.15–−23.19	−25.9 ± 1.79	MgCrO_4_
MgHPO_4_:3H_2_O	−0.85–0.22		−0.34 ± 0.29	−0.79–0.11	−0.42 ± 0.27	−1.12–−0.28	−0.59 ± 0.34	MgHPO_4_:3H_2_O
Nesquehonite	−1.90–−0.85		−1.50 ± 0.23	−1.79–−1.14	−1.36 ± 0.19	−2.54–−1.79	−2.18 ± 0.26	MgCO_3_:3H_2_O
Periclase	−9.71–−8.15		−9.21 ± 0.39	−9.56–−8.30	−8.87 ± 0.35	−11.42–−9.89	−10.73 ± 0.55	MgO
Mg(OH)_2_	−6.93–−5.36		−6.43 ± 0.39	−6.77–−5.52	−6.08 ± 0.35	−8.64–−7.11	−7.95 ± 0.55	Mn(OH)_2_
Halite	−4.48–−3.57		−4.07 ± 0.29	−4.41–−3.46	−3.80 ± 0.26	−3.72–−3.48	−3.59 ± 0.09	NaCl
Mirabilite	−4.09–−2.31		−3.25 ± 0.54	−3.94–−2.17	−2.70 ± 0.53	−2.46–−2.15	−2.32 ± 0.12	Na_2_SO_4_:10H_2_O
Na_2_Cr_2_O_7_	−41.51–−29.11		−36.81 ± 3.09	−40.22–−27.03	−34.33 ± 3.66	−44.66–−36.59	−41.22 ± 2.95	Na_2_Cr_2_O_7_
Na_2_CrO_4_	−22.90–−15.37		−20.24 ± 1.91	−21.99–−14.07	−18.46 ± 2.28	−24.55–−19.91	−22.61 ± 1.71	Na_2_CrO_4_
Natron	−6.28–−3.51		−5.17 ± 0.75	−6.18–−3.21	−4.31 ± 0.83	−4.91–−4.36	−4.69 ± 0.21	Na_2_CO_3_:10H_2_O
Thenardite	−5.47–−3.68		−4.64 ± 0.54	−5.33–−3.53	−4.07 ± 0.54	−3.82–−3.49	−3.67 ± 0.13	Na_2_SO_4_
Thermonatrite	−8.18–−5.39		−7.08 ± 0.76	−8.08–−5.10	−6.20 ± 0.84	−6.79–−6.25	−6.57 ± 0.20	Na_2_CO_3_:H_2_O
K_2_Cr_2_O_7_	−35.30–−24.19		−31.16 ± 2.83	−34.89–−22.24	−29.25 ± 3.36	−40.17–−31.65	−36.44 ± 3.11	K_2_Cr_2_O_7_
K_2_CrO_4_	−20.72–−14.35		−18.49 ± 1.62	−20.59–−13.18	−17.28 ± 1.97	−23.97–−18.87	−21.74 ± 1.87	K_2_CrO_4_
Cr(OH)_2_	−17.12–−14.80		−15.58 ± 0.64	−17.40–−15.01	−16.03 ± 0.75	−15.42–−14.80	−15.12 ± 0.22	Cr(OH)_2_
Cr(OH)_3_	2.80	5.56	4.46 ± 0.73	2.90–5.74	4.29 ± 0.84	3.70–5.02	4.29±0.48	Cr(OH)_3_
Cr_2_O_3_	6.47	12.00	9.79 ± 1.46	6.66–12.36	9.46 ± 1.68	8.28–10.92	9.46±0.95	Cr2O_3_
CrCl_2_	−37.53	−34.87	−35.88 ± 0.67	−38.72–−35.33	−36.86 ± 0.81	−34.60–−33.73	−34.12±0.31	CrCl_2_
CrCl_3_	−38.68	−35.87	−36.95 ± 0.68	−39.73–−36.36	−37.90 ± 0.81	−35.64–−34.73	−35.16±0.32	CrCl_3_
Cr-metal	−59.56–−57.16		−58.08 ± 0.66	−61.05–−57.61	−59.10 ± 0.82	−56.85–−56.06	−56.37 ± 0.29	Cr
CrO_3_	−28.22–−23.36		−26.19 ± 1.22	−27.85–−22.58	−25.49 ± 1.40	−29.72–−26.30	−28.23 ± 1.24	CrO_3_
Bunsenite	−3.06–−1.44		−2.38 ± 0.34	−3.62–−1.19	−2.29 ± 0.53	−5.15–−3.60	−4.48 ± 0.56	NiO
Morenosite	−4.76–−3.95		−4.24 ± 0.22	−6.10–−4.01	−4.66 ± 0.56	−5.11–−4.98	−5.05 ± 0.05	NiSO_4_:7H_2_O
Ni(OH)_2_	−3.42–−1.80		−2.74 ± 0.34	−3.98–−1.54	−2.64 ± 0.53	−5.50–−3.96	−4.84 ± 0.56	Ni(OH)_2_
Ni_3_(PO_4_)_2_	−2.14–1.08		−0.24 ± 0.80	−5.29–0.90	−0.80 ± 1.64	−5.65–−2.49	−3.99 ± 1.17	Ni_3_(PO_4_)_2_
Ni_4_(OH)_6_SO_4_	−10.58–−5.49		−8.17 ± 1.02	−13.74–−4.86	−8.31 ± 1.96	−17.31–−12.62	−15.27 ± 1.70	Ni_4_(OH)_6_SO_4_
NiCO_3_	−2.03–−1.01		−1.59 ± 0.21	−3.03–−0.98	−1.70 ± 0.48	−3.18–−2.42	−2.85 ± 0.27	NiCO_3_
Retgersite	−4.86–−4.05		−4.34 ± 0.22	−6.20–−4.11	−4.76 ± 0.56	−5.21–−5.08	−5.15 ± 0.05	NiSO_4_:6H_2_O
Hausmannite	−14.14–−5.78		−10.01 ± 2.10	−10.88–−4.56	−7.68 ± 1.84	−15.38–−10.96	−13.84 ± 1.71	Mn_3_O_4_
Birnessite	−12.75–−9.03		−11.21 ± 0.96	−11.66–−8.36	−10.05 ± 0.96	−14.23–−11.82	−13.31 ± 0.89	MnO_2_
Bixbyite	−11.27–−5.23		−8.43 ± 1.53	−9.09–−4.28	−6.68 ± 1.40	−12.62–−9.21	−11.40 ± 1.30	Mn_2_O_3_
Manganite	−5.95–−2.93		−4.53 ± 0.76	−4.86–−2.46	−3.66 ± 0.70	−6.63–−4.92	−6.01 ± 0.65	MnOOH
Mn_2_(SO_4_)_3_	−54.79–−51.05		−52.59 ± 0.70	−52.81–−51.49	−52.40 ± 0.37	−52.14–−50.98	−51.65 ± 0.56	Mn_2_(SO_4_)_3_
Mn_3_(PO_4_)_2_	−13.11–−7.57		−9.72 ± 1.14	−9.98–−6.90	−8.81 ± 0.71	−10.39–−8.81	−9.75 ± 0.67	Mn_3_(PO_4_)_2_
MnCl_2_:4H_2_O	−11.4–−9.61		−10.38 ± 0.36	−10.66–−9.70	−10.32 ± 0.21	−10.17–−9.58	−9.94 ± 0.25	MnCl_2_:4H_2_O
MnHPO_4_	2.88–−4.63		3.93 ± 0.37	3.70–4.78	4.09 ± 0.27	4.14–4.48	4.34 ± 0.16	MnHPO_4_
MnSO_4_	−10.68–−8.82		−9.60 ± 0.35	−9.76–−9.05	−9.53 ± 0.19	−9.40–−8.84	−9.16 ± 0.27	MnSO_4_
Nsutite	−12.16–−8.44		−10.62 ± 0.96	−11.07–−7.77	−9.46 ± 0.96	−13.64–−11.23	−12.73 ± 0.89	MnO_2_
Pyrochroite	−7.10–−4.65		−5.81 ± 0.58	−6.01–−4.51	−5.22 ± 0.45	−6.98–−5.97	−6.67 ± 0.42	Mn(OH)_2_
Pyrolusite	−10.69–−6.97		−9.15 ± 0.96	−9.60–−6.30	−7.99 ± 0.96	−12.17–−9.76	−11.25 ± 0.89	MnO_2_
Rhodochrosite	0.16–2.41		1.45 ± 0.49	1.22–2.27	1.83 ± 0.30	1.20–1.67	1.43 ± 0.23	MnCO_3_
O_2_(g)	−38.30–−35.10		−37.41 ± 0.87	−37.90–−34.30	−36.26 ± 1.05	−41.11–−38.30	−39.91 ± 1.02	O_2_
CH_4_(g)	−73.88–−66.72		−68.85 ± 1.90	−75.65–−67.81	−71.36 ± 2.28	−66.68–−60.28	−63.02 ± 2.33	CH_4_
CO_2_(g)	−0.79–−0.03		−0.38 ± 0.18	−0.96–−0.32	−0.59 ± 0.18	0.01–0.80	0.47 ± 0.29	CO_2_
Ionic strength	0.34–0.69		0.52 ± 0.25	0.43–0.66	0.55 ± 0.16	0.56–0.68	0.62 ± 0.08	
Total carbon	0.217–0.458		0.3373 ± 0.171	0.229–0.411	0.32 ± 0.13	0.515–0.585	0.55 ± 0.05	
Total CO_2_	0.217–0.458		0.3373 ± 0.171	0.23–0.411	0.32 ± 0.13	0.515–0.585	0.55 ± 0.05	
Electrical balance	−0.48–−0.35		−0.41 ± 0.091	−0.51–−0.44	−0.48 ± 0.05	−0.55–−0.5	−0.53 ± 0.03	
Percent error	−71.3–−44.5		−57.87 ± 18.92	−84.1–−41.9	−62.98 ± 29.9	−56.3–−50.3	−53.3 ± 4.19	
Iterations	10.0–12.0		11.0 ± 1.414	11.0–12.0	11.5 ± 0.71	11.0–12.0	11.5 ± 0.71	
Total H	111.2–111.5		111.36 ± 0.156	111.3–111.4	111.3 ± 0.11	111.6–111.8	111.7 ± 0.15	
Total O	56.59–57.85		57.22 ± 0.888	56.79–57.63	57.21 ± 0.59	57.85–57.87	57.86 ± 0.01	

## Data Availability

Research data can be obtained from the corresponding author through email.

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
