# Peer review of "Groundwater Quality, Health Risk Assessment, and Source Distribution of Heavy Metals Contamination around Chromite Mines: Application of GIS, Sustainable Groundwater Management, Geostatistics, PCAMLR, and PMF Receptor Model"

_ijerph, 2023, doi:10.3390/ijerph20032113_

Round 1
Reviewer 1 Report
General comments
The paper fits well the scope of the special issue and journal. The research is also robust. Despite this, there are several comments on the organization of the manuscript that needs to be fixed before publication
Specific comments
Lines 22-45. You need to drop some redundant detail in the abstract, leave space, and highlight the novelty of your research here
Lines 115-166. “It's a valuable tool for groundwater spatial distribution and autocorrelation as well”. Add relevant and recent references on the importance of a geospatial approach on groundwater protection. See below
- Medici, G. and Langman, J.B., 2022. Pathways and Estimate of Aquifer Recharge in a Flood Basalt Terrain; A Review from the South Fork Palouse River Basin (Columbia River Plateau, USA). Sustainability, 14(18), p.11349.
- Ducci, D. and Sellerino, M., 2013. Vulnerability mapping of groundwater contamination based on 3D lithostratigraphical models of porous aquifers. Science of the Total Environment, 447, pp.315-322.
Lines 50-147. The introduction is long, some detail can be moved to the next section
Line 147. Add the three specific objectives of your research using numbers (e.g., i, ii and iii)
Line 152. Issue of space and formatting after 35 degrees.
Lines 166-174. More details on the degree of humidity or aridity at the study site.
Line 218. Unclear the word “viz”
Line 248. You do need to say how you have plotted the data. From my side, you can drop the detail on Excel from the manuscript
Lines 298-299. More detail on the PHREEQC analysis, which chemical species did you treat? Insert reference to the USGS manual
Lines 298-299. You mention PHREEQC only in the methodology section. Recall the numerical code also in the results section when relevant
Lines 345-359. Too many figures/numbers. You need to find a way to drop from the main body this detail
Lines 440-446. Too many figures/numbers also here. You need to find a way to drop from the main body this detail
Lines 489-512. Too many figures/numbers also here. You need to find a way to drop from the main body this detail
Lines 568-588. Too many figures/numbers also here. You need to find a way to drop from the main body this detail
Lines 609-639. Too many figures/numbers also here. You need to find a way to drop from the main body this detail
Lines 853-865. Too many figures/numbers also here. You need to find a way to drop from the main body this detail
Line 899. “Hence, it” avoid to use “it” and repeat the subject. Please, fix this point here and throughout your manuscript.
Lines 924-929. Formatting issue. Please, fix it
Line 954. Add relevant references on geospatial analyses, aquifer vulnerability and groundwater quality that have been suggested above. One of the two paper is a review from an ultramafic terrain
Figures and tables
Figure 10. Insert labels also on the pie chart. So that people can read it also without colours
Author Response
Dear Editor and Reviewers,
We are really gratified for the time and importance honored to our manuscript. The reviewers studied this paper systematically and in detail to erase errors for improvement and the best of the scope, readability, and presentable manuscript. Collectively, the evaluations made it likely to improve our manuscript to a higher standard, and for this, we are very thankful to the team of the International Journal of Environmental Research and Public Health (editor and reviewers). We have tried our level best to incorporate all the individual points highlighted by the reviewers and revised the whole manuscript for a quality publication. The itemized replies are provided below along with the revised paper. We hope that the quality of this paper is improved and you find our responses to your satisfaction. We uploaded two versions of the revised paper, one with track changes and another clean (without track changes).
Sincerely,
Abdur Rashid 1,5,*, Muhammad Ayub 2, Zahid Ullah 1, Asmat Ali 1, Tariq Sardar 3, Javed Iqbal 4, Xubo Gao 1,*, Jochen Bundschuh 4, Chengcheng Li 1, Seema Anjum Khattak 5, Liaqat Ali 5, Hamed A. El-Serehy 6, Prashant Kaushik 7 and Sardar Khan 8
Reviewer # 1
Specific comments
Lines 22-45. You need to drop some redundant detail in the abstract, leave space, and highlight the novelty of your research here
Reply
Dear Editor and Reviewer, Thank you so much for your valuable comments, appreciation, and acceptance of our research work. We have modified the abstract by removing the unnecessary sentences and words as suggested by the Editor, and Reviewers.
Lines 115-166. “It's a valuable tool for groundwater spatial distribution and autocorrelation as well”. Add relevant and recent references on the importance of a geospatial approach on groundwater protection. See below
- Medici, G. and Langman, J.B., 2022. Pathways and Estimate of Aquifer Recharge in a Flood Basalt Terrain; A Review from the South Fork Palouse River Basin (Columbia River Plateau, USA). Sustainability, 14(18), p.11349.
- Ducci, D. and Sellerino, M., 2013. Vulnerability mapping of groundwater contamination based on 3D lithostratigraphical models of porous aquifers. Science of the Total Environment, 447, pp.315-322.
Reply
Dear Editor and Reviewer, Thank you so much for this important comment. We have added recent and suggested references in the revised manuscript as suggested by the Reviewer. Now we hope that the Editor and Reviewer can see the revised manuscript to their satisfaction. Here, the Reviewer can see the changes below.
“Medici, G., & Langman, J. B. (2022). Pathways and Estimate of Aquifer Recharge in a Flood Basalt Terrain; A Review from the South Fork Palouse River Basin (Columbia River Plateau, USA). Sustainability, 14(18), 11349.
Ducci, D., & Sellerino, M. (2013). Vulnerability mapping of groundwater contamination based on 3D lithostratigraphical models of porous aquifers. Science of the Total Environment, 447, 315-322.”
Lines 50-147. The introduction is long, some detail can be moved to the next section
Reply
Dear Editor and Reviewer, Thank you so much for this important comment. We have removed the unnecessary sentence from introduction part of the manuscript as suggested by the reviewer. However, Reviewer #2 also suggested some changes in the introduction section which are important information regarding HMs enrichment, transformation, and water management problems. Thus, Introduction will be modified accordingly. Now we hope that the Editor and Reviewer can see the revised manuscript to their satisfaction.
Line 147. Add the three specific objectives of your research using numbers (e.g., i, ii and iii)
Reply
Dear Editor and Reviewer, Thank you so much for this crucial comment. We have added numbers for the three specific objectives as suggested by the Editor, and Reviewer in the revised manuscript. Now we hope that the Editor and Reviewer can see the revised manuscript to their satisfaction.
Line 152. Issue of space and formatting after 35 degrees.
Reply
Dear Editor and Reviewer, Thank you so much for this crucial comment. We have corrected the mistakes regarding the 35 degree highlighted by the Reviewer in the revised manuscript. Now we hope that the Editor and Reviewer can see the changes to their satisfaction. Here, the Reviewer can see the changes below.
“District Malakand is an important area of Khyber Pakhtunkhwa province that lies between 34-35O Northing and 71-72O Easting, and occupied a 952 Km2 area (Figure 1).”
Lines 166-174. More details on the degree of humidity or aridity at the study site.
Reply
Dear Editor and Reviewer, Thank you so much for this important comment. We have added humidity or aridity of the area in the corrected manuscript. Now we hope that the Editor and Reviewer can see the changes to their satisfaction. Here, the Reviewer can see the changes below.
“The climate of the area is typically semiarid; with bimodal rainfall that peaks in spring and winter season with average annual precipitation of 950 mm. The research area experiences tremendously hot summers and severe cold in winter. Furthermore, the majority of the water sources are dry during hot summers due to severe arid conditions [68]. The yearly temperature ranges are 18.2 and 36.8 OC during summer and -6 to -16 OC in winter season [69]. On the other side; extreme seasonal change in perceived humidity is observed in the study area. It describes the amount of water vapor present in the atmosphere. Humidity increases with rise in temperature due to which water evaporate into atmosphere. The weather of the area is hot with more humidity due to high temperature and fast evaporation rate. Moreover, relative humidity calculates water molecules in the atmosphere in relation to maximum moisture content in air. The high and lower level of humidity is dangerous for young and old age people because of their high health risk exposure.”
“68. Iyyapazham, S., Managing water resources in agriculture and watersheds: modeling using GIS and dynamic simulation. University of Massachusetts Amherst: 2007.
- Grönwall, J. T.; Mulenga, M.; McGranahan, G., Groundwater, self-supply and poor urban dwellers: A review with case studies of Bangalore and Lusaka. IIED: 2010.”
Line 218. Unclear the word “viz”
Reply
Dear Editor and Reviewer, Thank you so much for this comment. We have modified the word viz. and replaced with “such as” in the corrected manuscript. Now we hope that the Editor and Reviewer can see the changes to their satisfaction.
Line 248. You do need to say how you have plotted the data. From my side, you can drop the detail on Excel from the manuscript.
Reply
Dear Editor and Reviewer, Thank you so much for this important comment. We have added the methodology for Gibbs and Chadha diagram in the revised manuscript. Now we hope that the Editor and Reviewer can see the changes to their satisfaction. Here, the Reviewer can see the changes below.
“Gibbs plot were calculated by plotting Log TDS against (Na/Na + Ca) and (Cl/Cl + HCO3). Whereas, Chadha diagram were statistically plotted between HCO3 - (Cl-SO4) against (Ca+Mg) - (Na+K) using their computation formula through a probabilistic approach.”
Lines 298-299. More detail on the PHREEQC analysis, which chemical species did you treat? Insert reference to the USGS manual
Reply
Dear Editor and Reviewer, Thank you so much for this comment. We have added the details of chemical analysis using PHREEQC interactive software for mineral phases and saturation indices in the revised manuscript. Now we hope that the Editor and Reviewer can see the changes to their satisfaction. Here, the Reviewer can see the changes below.
“The PHREEQC was used to measure mineral phases for physicochemical variables and HMs such as Cr, Ni, and Mn by following Eq. 7.
Parkhurst, D. L. (1995). User's guide to PHREEQC: A computer program for speciation, reaction-path, advective-transport, and inverse geochemical calculations (Vol. 95, No. 4227). US Department of the Interior, US Geological Survey.
Kinniburgh, D., & Cooper, D. (2011). PhreePlot: Creating graphical output with PHREEQC.”
Lines 298-299. You mention PHREEQC only in the methodology section. Recall the numerical code also in the results section when relevant.
Reply
Dear Editor and Reviewer, Thank you so much for this comment. We have added PHREEQC code in the result section in the revised manuscript. Now we hope that the Editor and Reviewer can see the changes to their satisfaction.
Lines 345-359. Too many figures/numbers. You need to find a way to drop from the main body this detail
Reply
Dear Editor and Reviewer, Thank you so much for this important comment. We have removed the unnecessary sentences and wording in the revised manuscript as suggested by the Editor, and Reviewer. Now we hope that the Editor and Reviewer can see the changes to their satisfaction.
Lines 440-446. Too many figures/numbers also here. You need to find a way to drop from the main body this detail
Reply
Dear Editor and Reviewer, Thank you so much for this comment. We have deleted the unnecessary words and modified the figure captions as suggested by the Editor, and Reviewer in the revised manuscript. Now we hope that the Editor and Reviewer can see the changes to their satisfaction.
Lines 489-512. Too many figures/numbers also here. You need to find a way to drop from the main body this detail
Reply
Dear Editor and Reviewer, Thank you so much for this important comment. We have removed the unnecessary sentences and words in the revised manuscript as suggested by the Editor, and Reviewer. Now we hope that the Editor and Reviewer can see the changes to their satisfaction.
Lines 568-588. Too many figures/numbers also here. You need to find a way to drop from the main body this detail
Reply
Dear Editor and Reviewer, Thank you so much for this important comment. We have removed the unnecessary sentences and words as suggested by the Editor, and Reviewer from our corrected manuscript. Now we hope that the Editor and Reviewer can see the changes to their satisfaction.
Lines 609-639. Too many figures/numbers also here. You need to find a way to drop from the main body this detail
Reply
Dear Editor and Reviewer, Thank you so much for this important comment. We have deleted the unnecessary information’s in the revised manuscript. Now we hope that the Editor and Reviewer can see the changes to their satisfaction.
Lines 853-865. Too many figures/numbers also here. You need to find a way to drop from the main body this detail
Reply
Dear Editor and Reviewer, Thank you so much for this important comment. We have removed the unnecessary sentences in the revised manuscript as suggested by the Editor, and Reviewer. Now we hope that the Editor and Reviewer can see the changes to their satisfaction.
Line 899. “Hence, it” avoid to use “it” and repeat the subject. Please, fix this point here and throughout your manuscript.
Reply
Dear Editor and Reviewer, Thank you so much for this crucial comment. We have added the subject instead of “it” in the suggested place and further corrected throughout the manuscript. Now we hope that the Editor and Reviewer can see the changes to their satisfaction.
Lines 924-929. Formatting issue. Please, fix it
Reply
Dear Editor and Reviewer, Thank you so much for this important comment. We have fixed the formatting issues in the suggested place as recommended by Editor, and Reviewer. Now we hope that the Editor and Reviewer can see the changes to their satisfaction.
Line 954. Add relevant references on geospatial analyses, aquifer vulnerability and groundwater quality that have been suggested above. One of the two paper is a review from an ultramafic terrain
Reply
Dear Editor and Reviewer, Thank you so much for this this comment. We have added relevant references for geospatial analyses, aquifer vulnerability and groundwater quality as recommended by Editor, and Reviewer. Now we hope that the Editor and Reviewer can see the changes to their satisfaction.
Figures and tables
Figure 10. Insert labels also on the pie chart. So that people can read it also without colours
Reply
Dear Editor and Reviewer, Thank you so much for your comment. Actually, we used EPAPMF software for Figure 10, the labeling on the pie chart of each element is quite difficult/possible disable in the software. We have done our level best to improve the quality of the revised manuscript as suggested by the Editor, and Reviewer. Now we hope that the Editor and Reviewer can see the revised manuscript to their satisfaction.
Thank You so much.
Regards
Abdur Rashid corresponding author

Reviewer 2 Report
This article systematically introduces the spatial distribution, water quality, geochemical composition, groundwater management, and health impacts of heavy metals in groundwater around the enriched chromite mines. This is very important for local environmental management. However, the authors need to do much more to improve the paper and make a good scientific story. Kindly, see below for detailed comments:
1 In the abstract, please highlight the outstanding contributions and guiding conclusions of this article.
2 Keywords, ‘Heavy metal’ should be added.
3 Introduction, this article introduces the harm of Cr in a large space. I suggest the focus should be on the migration, transformation, and enrichment process of heavy metals in groundwater brought about by the industrialization of urbanization, and what are the problems faced by groundwater management.
4 Heat Map Construction, I cannot get the meaning of heat map in this article.
5 Results and Discussion, the author has done a lot of work, but the article should not be written like this. It is necessary to summarize the rules of the results, find out the differences in the results, and analyze the reasons for the differences. Not every result should be explained in detail, and the results should be explained in short words. After the table is available, the numerical value of the text can be omitted, and only the important ones can be explained. The discussion also lacks the comparison of relevant research.
6 The Conclusion should be highlighting the significance and guiding role of this study.
7 I strongly suggest the author consider having a round of language checking and try to avoid long, complicated sentences to make this manuscript more readable.
Author Response
Dear Editor and Reviewers,
We are really gratified for the time and importance honored to our manuscript. The reviewers studied this paper systematically and in detail to erase errors for improvement and the best of the scope, readability, and presentable manuscript. Collectively, the evaluations made it likely to improve our manuscript to a higher standard, and for this, we are very thankful to the team of the International Journal of Environmental Research and Public Health (editor and reviewers). We have tried our level best to incorporate all the individual points highlighted by the reviewers and revised the whole manuscript for a quality publication. The itemized replies are provided below along with the revised paper. We hope that the quality of this paper is improved and you find our responses to your satisfaction. We uploaded two versions of the revised paper, one with track changes and another clean (without track changes).
Sincerely,
Abdur Rashid 1,5,*, Muhammad Ayub 2, Zahid Ullah 1, Asmat Ali 1, Tariq Sardar 3, Javed Iqbal 4, Xubo Gao 1,*, Jochen Bundschuh 4, Chengcheng Li 1, Seema Anjum Khattak 5, Liaqat Ali 5, Hamed A. El-Serehy 6, Prashant Kaushik 7 and Sardar Khan 8
Reviewer #2
This article systematically introduces the spatial distribution, water quality, geochemical composition, groundwater management, and health impacts of heavy metals in groundwater around the enriched chromite mines. This is very important for local environmental management. However, the authors need to do much more to improve the paper and make a good scientific story. Kindly, see below for detailed comments:
Reply
Dear Editor and Reviewer, Thank you so much for your positive comments, appreciation, and acceptance of our research work. We agreed with the policy of the Journal.
1 In the abstract, please highlight the outstanding contributions and guiding conclusions of this article.
Reply
Dear Editor and Reviewer, Thank you so much for your valuable comments. We have modified the abstract as suggested by the Editor, and Reviewers. Now we hope that the Editor and Reviewer can see the revised manuscript to their satisfaction.
2 Keywords, ‘Heavy metal’ should be added.
Reply
Dear Editor and Reviewer, Thank you so much for this comment. We have added heavy metals in the keywords as suggested by the Editor, and Reviewer. Now we hope that the Editor and Reviewer can see the revised manuscript to their satisfaction.
3 Introduction, this article introduces the harm of Cr in a large space. I suggest the focus should be on the migration, transformation, and enrichment process of heavy metals in groundwater brought about by the industrialization of urbanization, and what are the problems faced by groundwater management.
Reply
Dear Editor and Reviewer, Thank you so much for this comment. We have tried our level best to modify the introduction section of the revised manuscript by adding the literature about the migration, transformation, and enrichment process of heavy metals in groundwater systems brought about by the industrialization of urbanization, and associated problems faced by groundwater management in the study area. Now we hope that the Editor and Reviewer can see the revised manuscript to their satisfaction. Here, the Reviewer can see the changes below.
“The continuous demand for groundwater supply increases due to industrialization, urbanization, chromite mining, and the growing population. However, migration, transformation, and enrichment of HMs in groundwater systems mostly depend on the structural composition of groundwater aquifer and subsurface soil [54, 55]. Moreover, the transformation of HMs in groundwater depends on the composition of aquifer media, types, and concentrations [56]. The role of microbes such as bacteria, and fungi are effective in the transformation of HMs from one form into another form [57]. HMs released into the groundwater system; if the HMs retaining capacity of the soil subsurface decrease due to pH fluctuation [58, 59]. Many areas of Pakistan are experiencing groundwater deterioration [3, 60]. Therefore, effective groundwater management is essential to maintain the long-term survival of groundwater aquifers [61, 62]. However, poorly planned land development strategies deteriorate water quality in Pakistan [63]. In Pakistan, the lack of capacity of groundwater resource managers, and coordination between concerned stakeholders such as government institutions, communities, and the private sector is inadequate [64, 65]. The national law allowed local governments to design effective groundwater management plans [66], to address issues including groundwater contamination, wellhead protection, groundwater remediation, over-extraction, overdraft, recharge, improper irrigation, seawater intrusion, water storage, and conservation [67-71]. Groundwater is becoming less available and less safe to use as a result of pollution, over-extraction, and removal techniques [72, 73]. Therefore, improved groundwater management is crucial for a healthy, and green Pakistan [74].”
“53. Zhang, L.; Qin, X.; Tang, J.; Liu, W.; Yang, H., Review of arsenic geochemical characteristics and its significance on arsenic pollution studies in karst groundwater, Southwest China. Applied geochemistry 2017, 77, 80-88.
- Nikolenko, O.; Jurado, A.; Borges, A. V.; Knӧller, K.; Brouyѐre, S., Isotopic composition of nitrogen species in groundwater under agricultural areas: a review. Science of the Total Environment 2018, 621, 1415-1432.
- Ogundiran, M.; Osibanjo, O., Mobility and speciation of heavy metals in soils impacted by hazardous waste. Chemical Speciation & Bioavailability 2009, 21, (2), 59-69.
- Thakare, M.; Sarma, H.; Datar, S.; Roy, A.; Pawar, P.; Gupta, K.; Pandit, S.; Prasad, R., Understanding the holistic approach to plant-microbe remediation technologies for removing heavy metals and radionuclides from soil. Current Research in Biotechnology 2021, 3, 84-98.
- Tauqeer, H. M.; Fatima, M.; Rashid, A.; Shahbaz, A. K.; Ramzani, P. M. A.; Farhad, M.; Basharat, Z.; Turan, V.; Iqbal, M., The current scenario and prospects of immobilization remediation technique for the management of heavy metals contaminated soils. In Approaches to the remediation of inorganic pollutants, Springer: 2021; pp 155-185.
- Kumar, M.; Kushwaha, A.; Goswami, L.; Singh, A. K.; Sikandar, M., A review on advances and mechanism for the phycoremediation of cadmium contaminated wastewater. Cleaner Engineering and Technology 2021, 5, 100288.
- Rashid, A.; Ayub, M.; Javed, A.; Khan, S.; Gao, X.; Li, C.; Ullah, Z.; Sardar, T.; Muhammad, J.; Nazneen, S., Potentially harmful metals, and health risk evaluation in groundwater of Mardan, Pakistan: Application of geostatistical approach and geographic information system. Geoscience Frontiers 2021, 12, (3), 101128.
- Rashid, A.; Ayub, M.; Ullah, Z.; Ali, A.; Khattak, S. A.; Ali, L.; Gao, X.; Li, C.; Khan, S.; El-Serehy, H. A., Geochemical Modeling Source Provenance, Public Health Exposure, and Evaluating Potentially Harmful Elements in Groundwater: Statistical and Human Health Risk Assessment (HHRA). International Journal of Environmental Research and Public Health 2022, 19, (11), 6472.
- Foster, S.; Chilton, J.; Nijsten, G.-J.; Richts, A., Groundwater—a global focus on the ‘local resource’. Current opinion in environmental sustainability 2013, 5, (6), 685-695.
- Gleeson, T.; Richter, B., How much groundwater can we pump and protect environmental flows through time? Presumptive standards for conjunctive management of aquifers and rivers. River research and applications 2018, 34, (1), 83-92.
- Khalid, S., An assessment of groundwater quality for irrigation and drinking purposes around brick kilns in three districts of Balochistan province, Pakistan, through water quality index and multivariate statistical approaches. Journal of Geochemical Exploration 2019, 197, 14-26.
- Qureshi, A. S., Groundwater governance in Pakistan: From colossal development to neglected management. Water 2020, 12, (11), 3017.
- Ngene, B. U.; Nwafor, C. O.; Bamigboye, G. O.; Ogbiye, A. S.; Ogundare, J. O.; Akpan, V. E., Assessment of water resources development and exploitation in Nigeria: A review of integrated water resources management approach. Heliyon 2021, 7, (1), e05955.
- Basharat, M., Water management in the Indus Basin in Pakistan: challenges and opportunities. Indus River Basin 2019, 375-388.
- Mohan, S.; Kuipally, N., Groundwater and Conjunctive Use Management. In Handbook of Water Resources Management: Discourses, Concepts and Examples, Springer: 2021; pp 735-775.
- Iyyapazham, S., Managing water resources in agriculture and watersheds: modeling using GIS and dynamic simulation. University of Massachusetts Amherst: 2007.
- Grönwall, J. T.; Mulenga, M.; McGranahan, G., Groundwater, self-supply and poor urban dwellers: A review with case studies of Bangalore and Lusaka. IIED: 2010.
- Talpur, S. A.; Noonari, T. M.; Rashid, A.; Ahmed, A.; Jat Baloch, M. Y.; Talpur, H. A.; Soomro, M. H., Hydrogeochemical signatures and suitability assessment of groundwater with elevated fluoride in unconfined aquifers Badin district, Sindh, Pakistan. SN Applied Sciences 2020, 2, (6), 1-15.
- Iqbal, J.; Su, C.; Rashid, A.; Yang, N.; Baloch, M. Y. J.; Talpur, S. A.; Ullah, Z.; Rahman, G.; Rahman, N. U.; Sajjad, M. M., Hydrogeochemical assessment of groundwater and suitability analysis for domestic and agricultural utility in Southern Punjab, Pakistan. Water 2021, 13, (24), 3589.
- Gorelick, S. M.; Zheng, C., Global change and the groundwater management challenge. Water Resources Research 2015, 51, (5), 3031-3051.
- Habib, Z., Water availability, use and challenges in Pakistan-Water sector challenges in the Indus Basin and impact of climate change. Food & Agriculture Org.: 2021.
- Noor, S.; Rashid, A.; Javed, A.; Khattak, J. A.; Farooqi, A., Hydrogeological properties, sources provenance, and health risk exposure of fluoride in the groundwater of Batkhela, Pakistan. Environmental Technology & Innovation 2022, 25, 102239.”
4 Heat Map Construction, I cannot get the meaning of heat map in this article.
Reply
Dear Editor and Reviewer, Thank you so much for this important comment. We have deleted the heat map from the revised manuscript to better understand the remaining results of this research paper. Now we hope that the Editor and Reviewer can see the revised manuscript to their satisfaction.
5 Results and Discussion, the author has done a lot of work, but the article should not be written like this. It is necessary to summarize the rules of the results, find out the differences in the results, and analyze the reasons for the differences. Not every result should be explained in detail, and the results should be explained in short words. After the table is available, the numerical value of the text can be omitted, and only the important ones can be explained. The discussion also lacks the comparison of relevant research.
Reply
Dear Editor and Reviewer, Thank you so much for this important comment. We have modified the Results and discussion section of the revised manuscript as suggested by the Reviewer. Now we hope that the Editor and Reviewer can see the revised manuscript to their satisfaction.
6 The Conclusion should be highlighting the significance and guiding role of this study.
Reply
Dear Editor and Reviewer, Thank you so much for this crucial comment. We have modified the conclusion section of the corrected manuscript as suggested by the Reviewer. Now we hope that the Editor and Reviewer can see the revised manuscript to their satisfaction.
7 I strongly suggest the author consider having a round of language checking and try to avoid long, complicated sentences to make this manuscript more readable.
Reply
Dear Editor and Reviewer, Thank you so much for this comment. We have corrected the language of the revised manuscript by taking help from a native English speaker as suggested by the Reviewer. Now we hope that the Editor and Reviewer can see the revised manuscript to their satisfaction.
Thank You so much.
Regards
Abdur Rashid corresponding author

Round 2
Reviewer 2 Report
The revised manuscript has been greatly revised compared with the original manuscript, which has met the publishing requirements of IJERPH, but I still suggest that the results should be simplified, with fewer results listed and more results summarized.
Author Response
Dear Editor and Reviewers,
We are really gratified for the time and importance honored to our manuscript. The reviewers studied this paper systematically and in detail to erase errors for improvement and the best of the scope, readability, and presentable manuscript. Collectively, the evaluations made it likely to improve our manuscript to a higher standard, and for this, we are very thankful to the team of the International Journal of Environmental Research and Public Health (editor and reviewers). We have tried our level best to incorporate all the individual points highlighted by the reviewers and revised the whole manuscript for a quality publication. The itemized replies are provided below along with the revised paper. We hope that the quality of this paper is improved and you find our responses to your satisfaction. We uploaded two versions of the revised paper, one with track changes and another clean (without track changes).
Sincerely,
Abdur Rashid 1,5,*, Muhammad Ayub 2, Zahid Ullah 1, Asmat Ali 1, Tariq Sardar 3, Javed Iqbal 4, Xubo Gao 1,*, Jochen Bundschuh 4, Chengcheng Li 1, Seema Anjum Khattak 5, Liaqat Ali 5, Hamed A. El-Serehy 6, Prashant Kaushik 7 and Sardar Khan 8
Comments and Suggestions for Authors
The revised manuscript has been greatly revised compared with the original manuscript, which has met the publishing requirements of IJERPH, but I still suggest that the results should be simplified, with fewer results listed and more results summarized.
Reply
Dear Editor and Reviewer, Thank you so much for your positive comments, appreciation, and acceptance of our research work. We have remodified the revised manuscript and corrected the results according to the suggestion of the Editor, and Reviewers. Now we hope that the Editor and Reviewer can see the corrected manuscript to their satisfaction.
Thank You so much.
Regards
Abdur Rashid corresponding author